# Anionic redox reaction in layered $NaCr_{2/3}Ti_{1/3}S_2$ through electron holes formation and dimerization of S–S

Tian Wang [1,12], Guo-Xi Ren[2,3,4,12], Zulipiya Shadike[5,12], Ji-Li Yue[6], Ming-Hui Cao[1], Jie-Nan Zhang [7], Ming-Wei Chen[8], Xiao-Qing Yang [5], Seong-Min Bak[5], Paul Northrup[9], Pan Liu[8]*, Xiao-Song Liu[2,4,10,11]* & Zheng-Wen Fu[1]*

The use of anion redox reactions is gaining interest for increasing rechargeable capacities in alkaline ion batteries. Although anion redox coupling of $S^{2-}$ and $(S_2)^{2-}$ through dimerization of S–S in sulfides have been studied and reported, an anion redox process through electron hole formation has not been investigated to the best of our knowledge. Here, we report an O3-$NaCr_{2/3}Ti_{1/3}S_2$ cathode that delivers a high reversible capacity of ~186 mAh g$^{-1}$ (0.95 Na) based on the cation and anion redox process. Various charge compensation mechanisms of the sulfur anionic redox process in layered $NaCr_{2/3}Ti_{1/3}S_2$, which occur through the formation of disulfide-like species, the precipitation of elemental sulfur, S–S dimerization, and especially through the formation of electron holes, are investigated. Direct structural evidence for formation of electron holes and $(S_2)^{n-}$ species with shortened S–S distances is obtained. These results provide valuable information for the development of materials based on the anionic redox reaction.

[1] Shanghai Key Laboratory of Molecular Catalysts and Innovative Materials, Department of Chemistry & Laser Chemistry Institute, Fudan University, 200433 Shanghai, China. [2] State Key Laboratory of Functional Materials for Informatics, Shanghai Institute of Microsystem and Information Technology, Chinese Academy of Science, 200050 Shanghai, China. [3] University of Chinese Academy of Sciences, 100049 Beijing, China. [4] CAS Center for Excellence in Superconducting Electronics (CENSE), Chinese Academy of Sciences, 200050 Shanghai, China. [5] Chemistry Division, Brookhaven National Laboratory, Upton, NY 11973, USA. [6] School of Materials Science and Engineering, Herbert Gleiter Institute of Nanoscience, Nanjing University of Science and Technology, 210094 Nanjing, Jiangsu, China. [7] Beijing National Laboratory for Condensed Matter Physics, Institute of Physics, Chinese Academy of Sciences, 100190 Beijing, China. [8] Shanghai Key Laboratory of Advanced High-temperature Materials and Precision Forming, State Key Laboratory of Metal Matrix Composites, School of Materials Science and Engineering, Shanghai Jiao Tong University, 200240 Shanghai, China. [9] Department of Geosciences, Stony Brook University, Stony Brook, NY 11794, USA. [10] Tianmu Lake Institute of Advanced Energy Storage Technologies, 213300 Liyang City, Jiangsu, China. [11] School of Physical Science and Technology, ShanghaiTech University, 201210 Shanghai, China. [12] These authors contributed equally: Tian Wang, Guo-Xi Ren, Zulipiya Shadike. *email: panliu@sjtu.edu.cn; xliu3@mail.sim.ac.cn; zwfu@fudan.edu.cn

The discovery of anionic redox chemistry in layered compounds was traced back to ligand–hole chemistry in chalcogenides, in which $S_2^{2-}$, $Se_2^{2-}$, or $Te_2^{2-}$ ions in layered structures may be involved[1–6]. In recent years, attention is boosted by interest in anionic redox in oxides for high-energy Li-ion batteries[7–12]. These anionic redox-based oxides, such as layered $Li_{1.2}Ni_{0.13}Mn_{0.54}Co_{0.13}O_2$ (Li-rich NMC) provide nearly twice as much capacity as that of $LiCoO_2$ and $LiNi_{1/3}Mn_{1/3}Co_{1/3}O_2$ (NMC) cathodes[13–15]. The reversible activity of lattice oxygen in Li-rich NMC should be responsible for their extraordinary capacities[16,17]. In these compounds, the high electrochemical activity of O $2p$ non-bonding states is revealed[18–20]. The reversible formation of O–O dimers, as well as electron holes on O can also be confirmed in $Li_2IrO_3$, $Li_2RuO_3$, $Na_3RuO_4$[18,21,22], and $Li_{1.2}Ni_{0.13}Co_{0.13}Mn_{0.54}O_2$[17], respectively. Accordingly, the questions are whether these concepts could be mutually corroborated by other systems, such as layered chalcogenides, whether the reversible formation and decomposition of S–S dimers or electron holes on S could be evidenced, and what is the underlying nature of anionic redox chemistry in layered chalcogenides. The reversible anionic reduction/oxidation of $(S_2)^{2-} + 2e^- \leftrightarrow 2S^{2-}$ was reported previously in non-layered polysulfide electrode materials[23–28] and in layered materials confirmed by X-ray photoelectron spectroscopy (XPS) and X-ray absorption spectroscopy (XAS) techniques[29–31]. However, there has been a lack of core structural evidence on the formation of S–S dimers or electron holes on S in layered chalcogenides. In fact, the anionic redox activities from dimers or from electron holes is still controversial[17,32,33], which is found to be dependent on the composition and structure of the electrode. Here, we report direct structural evidences for the presence of $(S_2)^{n-}$ species with shortened S–S distances in the anionic redox processes in O3-$NaCr_{2/3}Ti_{1/3}S_2$ as a model layered cathode. The highly reversible formation and decomposition of S–S dimers driven by the migration of Cr ions are directly observed. The anionic redox chemistry of sulfur involving the formation of localized electron holes, anionic dimers, disulfide-like species, as well as the precipitation of elemental sulfur are in layered $NaCr_{2/3}Ti_{1/3}S_2$, in which the possible routes of anionic redox processes from the electron holes to element are clarified. A new pathway to engineer the anionic redox process for the optimal balance between the extra capacities boosted by $S^{2-}/S^-$ redox is explored.

## Results

**Enhanced capacity by anionic and cationic redox couples.** The members of $NaCr_{1-y}Ti_yS_2$ series were synthesized by high-temperature solid-state reactions using the mixture of powdered $Na_2S$, S, Ti, and Cr in various stoichiometric ratios. All the $NaCr_{1-y}Ti_yS_2$ series have O3 type structure, similar to $NaTiS_2$ and $NaCrS_2$ (the preliminary structural characterizations are shown in Supplementary Figs. 1, 2 and Supplementary Table 1). The electrochemical performance of $NaCr_{1-y}Ti_yS_2$ compounds was tested versus Na and cycled between 1.4 and 3.3 V (1.5–4.0 V for $NaCrS_2$) at the current rate of C/10 and the charge/discharge curves of the initial three cycles are shown in Fig. 1a. The capacity of pure $NaCrS_2$ comes mainly from the high voltage range above 2.55 V, corresponding to the redox between $S^{2-}$ and $S_2^{2-}$[31], with a characteristic of large polarization about 0.6 V. As contrast, the capacity of $NaTiS_2$ comes mainly from low voltage range below 2.55 V, corresponding to the redox of $Ti^{3+}$[34], with a small polarization about 0.05 V. These features can be observed in the curves of $NaCr_{1-y}Ti_yS_2$ series, which show small polarization in low voltage range and large polarization in high voltage range. The proportion of capacity from high/low voltage range relates to the ratio of Cr/Ti doped. Among the $NaCr_{1-y}Ti_yS_2$ series,

$NaCr_{2/3}Ti_{1/3}S_2$ shows the highest capacity of 186 mAh $g^{-1}$ in the second cycle, corresponding to the reversible deintercalation of 0.95 Na per unit. It can be found that three main stages in the charging process of $NaCr_{2/3}Ti_{1/3}S_2$ consist of the first plateau between 1.4 and 2.55 V, the slope between 2.55 and 2.85 V and the second plateau at 2.90 V (Fig. 1b). The capacity retentions with 69% of the first cycle is obtained after 50 cycles in the voltage range of 1.4–3.3 V, but the capacity remains unattenuated in the stage of 1.4–2.55 and 2.55–2.85 V (Fig. 1b inset). The capacity in the range of 1.4–2.55 V is 65 mAh $g^{-1}$, which conforms exactly to the $Ti^{3+/4+}$ fully redox. Both the stages of 2.55–2.85 and 2.85–3.3 V correspond to the redox processes of sulfur confirmed via in situ X-ray absorption near-edge structure (XANES) spectra (see the "Methods" section for XAS details), standing for the redox of $S^{2-}/S^{n-}$ and $S^{n-}/(S_2)^{n-}$, respectively. An intuitionistic pie chart representing the increase of capacity is shown in Supplementary Fig. 3. Significantly, more sulfur can be activated by the addition of Ti (33.33%) into $NaCrS_2$ to produce higher capacity. The cyclic voltammetry test (Fig. 1c) at a scan rate of 0.01 mV $s^{-1}$ shows three single anodic peaks at 1.7, 2.7, and 2.9 V standing for the oxidation processes of Ti/S/S, respectively, as mentioned above. Their corresponding three cathodic peaks locate at 1.6, 2.3, and 2.5 V, respectively. Rate performances were also tested between 0.1–1 and 0.1–10 C. The electrode showed good rate performance and could achieve the capacity of at least 80 mAh $g^{-1}$ even under such a high rate of 10 C (see Supplementary Fig. 4). After 360 cycles under 1/3 C rate, the capacity retention can reach 52.2% (see Supplementary Fig. 5). Comparing to other metal polysulfide electrodes[24–28] for lithium or sodium batteries (Supplementary Table 2), it is clearly shown that poor cycling stability is a common problem to be solved in metal sulfide electrodes driven by anion redox reactions, to which the highly reversibility of electron holes might be a potential solution as we discuss later.

For further insight into the structural changes of $NaCr_{2/3}Ti_{1/3}S_2$ electrode in desodiation processes, in situ XRD test was performed (Fig. 2a). The sodium deintercalation/intercalation process during the first cycle is highly reversible for the whole contour is symmetrical, and the structure after discharging to 1.4 V comes back to that of the pristine. Three corresponding stages in Fig. 1 can be further analyzed in Fig. 2a. The initial structure is O3 and the first stage corresponds to O3 → P3 transition. Then the solid solution reaction is occurred in P3 phase, which corresponds to the slope part in charge curve with shortening of $a/b$ lattice length and elongation along the $c$-axis. The third part is a P3-O1′ transition, which corresponds to the second plateau in Fig. 1b. A huge reduction in lattice parameter of $c$ occurred, implying the structural contraction after complete desodiation. The calculated cell parameters during the first charging process in Fig. 2b show that the lattice parameter of $c$, with considering the Cr migration, is closer to the experimental value. The $c$ length slightly decreased with the migration of Cr ions to Na vacancy sites because the electrostatic attraction between S layers becomes stronger.

**Explicit support of redox process via in situ/operando XAS.** The XAS is a powerful tool to directly provide the information of valence electrons in the vicinity of Fermi level, which is sensitive to the elements, chemical bonds, and oxidation states. Figure 3a–c show a complete in situ XAS dataset for Ti, Cr, and S during the charge and discharge process of $NaCr_{2/3}Ti_{1/3}S_2$/Na battery cycled at $C/10$. The in situ XAS evolve contours of three elements, exhibiting overall asymmetry, which shows the similarity to their corresponding charge and discharge profiles. Several features from three elements during the charge process are listed in the selected spectra shown in Fig. 3d–f, respectively. It can be seen

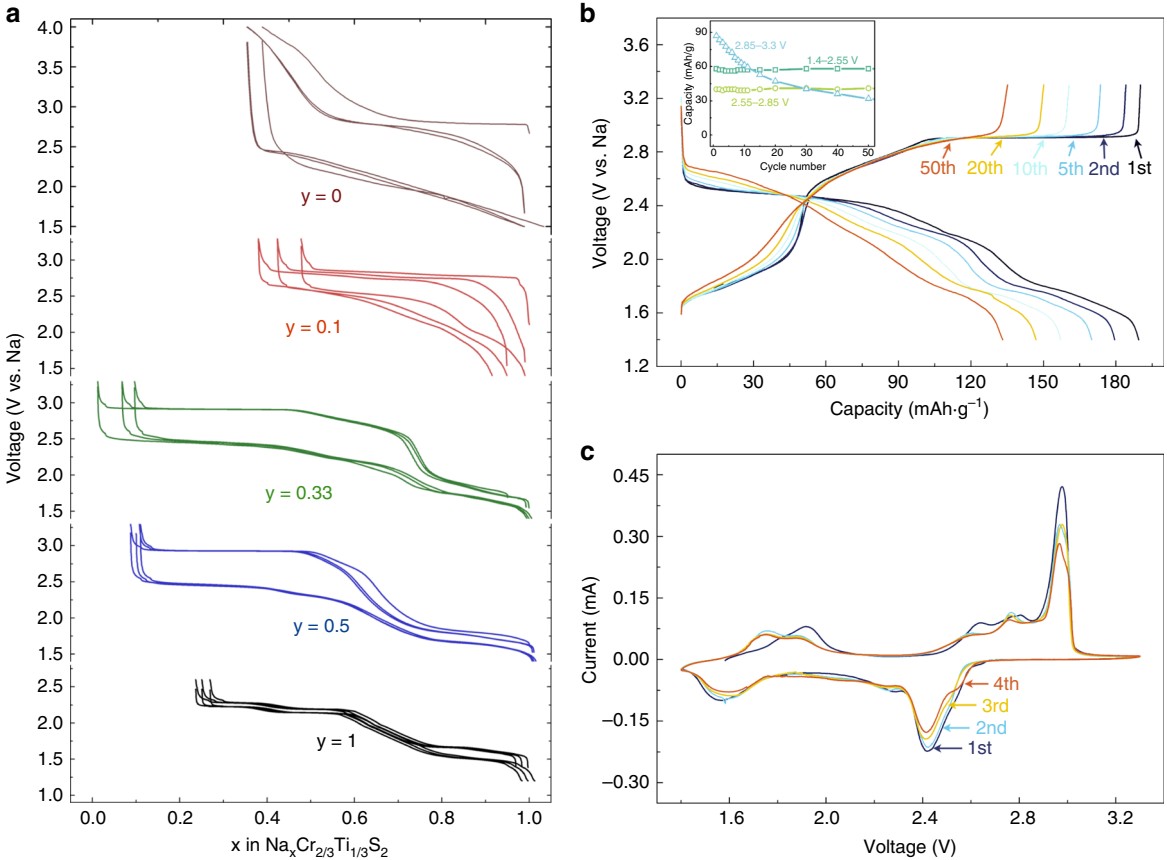

**Fig. 1** Electrochemical performance of $NaCr_{1-y}Ti_yS_2$ series. **a** Voltage profile for the $NaCr_{1-y}Ti_yS_2$ series versus $Na^+/Na$ at C/10 rate. The Coulombic efficiency of $NaCr_{2/3}Ti_{1/3}S_2$ at the first cycle is superior to 100% for there are few Na vacancies in the pristine material without Na precursor excess. **b** The voltage profile over first 50 cycles for a $Na/NaCr_{2/3}Ti_{1/3}S_2$ cell together with (as inset) its capacity retention in range of 1.4–2.55, 2.55–2.85 and 2.95–3.3 V at C/10 rate. **c** First four cyclic voltammograms for the $NaCr_{2/3}Ti_{1/3}S_2$ electrode cycled between 3.3 and 1.4 V

that during the whole charge process the Cr K-edge XAS is hardly changed, as its corresponding mapping result shows, indicating that Cr ions do not participate in the electrochemical redox reactions. Only a small rise in pre-edge, corresponding to the transition of Cr $1s$ electron to the $3d$-$t_{2g}$ and $3d$-$e_g$ orbitals through the hybridization of S-$3p$ and Cr-$3d$ states, is observed in stage 3 that might be resulting from the local distortion of Cr–S octahedron caused by movement of S. On the contrary, the Ti K-edge XAS spectra change in the first stage, the main peak located at ~4977.8 eV shifts continuously toward high energy, finally reaches 4978.8 eV and the pre-edge slightly rises due to holes generated on Ti$3d$S$3p$ partly admixed orbitals or due to the change of chemical environment between O3–P3 phase transition. The Na constant curve of Ti K-edge XAS spectra implies that there is no further oxidation reaction for Ti in subsequent stages and the oxidation reaction of $Ti^{3+}$ have been finished in the first stage. The most significant spectral variations at different state of charge are found in the XAS spectra at sulfur K-edge. First of all, during the first stage, the S K-edge is hardly changed. In the second stage, the shoulder located at ~2468 eV is gradually grown. Such a shoulder is attributed to the hybridization between the Cr $3d$ and S $3p$ orbitals. According to the structural evolution of solid solution P3 phase in this charging stage, the generation of electron holes on sulfur sites should be responsible for the formation of the typical shoulder, indicating the charge compensation taking place on sulfur sites in this desodiation process. The growth of a new peak appearing at 2470.7 eV can be clearly seen on the third stage of charging process, which is consistent with

the changes we have seen previously for $NaCrS_2$[31] and in previous report of $Li_2FeS_2$[35]. It corresponds to a newly formed localized electronic states on sulfur, probably stand for S–S $\sigma^*$, resulting from the occurrence of $2S^{2-}/(S_2)^{n-}$ ($n < 4$), i.e. the anionic redox process for charge compensation accompanied with the formation of S–S dimers[36,37]. The ex situ extended X-ray absorption fine structure (EXAFS) of sulfur was also collect after performing 2D X-ray fluorescence (XRF) and analyzed (see Fig. 3 bottom inset and Supplementary Figs. 6 and 7). An embedded peak raised at ~1.6 Å can be observed. This peak can be attributed to disulfide (S–S distance 2.05 Å) after considering phase correction, which is similar to the sulfur spectrum of $FeS_2$[38]. In addition, it should be noted that the line shape of S K-edge in XAS spectra varies systematically and its evolutions shows an isosbestic point upon desodiation process in the third stage, which indicates a dominating two-phase transformation[39], and is in good agreement with the XRD. Finally, like the evolution behavior of Cr K-edge during the charging process, the Cr $3d$-bands (see Supplementary Fig. 8) are almost inert to the discharging process. In the full discharging process, K-edge XAS spectra of Ti and S returns close to those of the pristine material in a symmetrical way, which indicates the reversibility of the charge compensation on Ti and S.

**Following the trail of S–S dimer formation in microstructure.** To further assess the reason behind the charge compensation during the first cycle, we tested the $NaCr_{2/3}Ti_{1/3}S_2$ electrode at three stages in charging process by XRD, selected-area electron

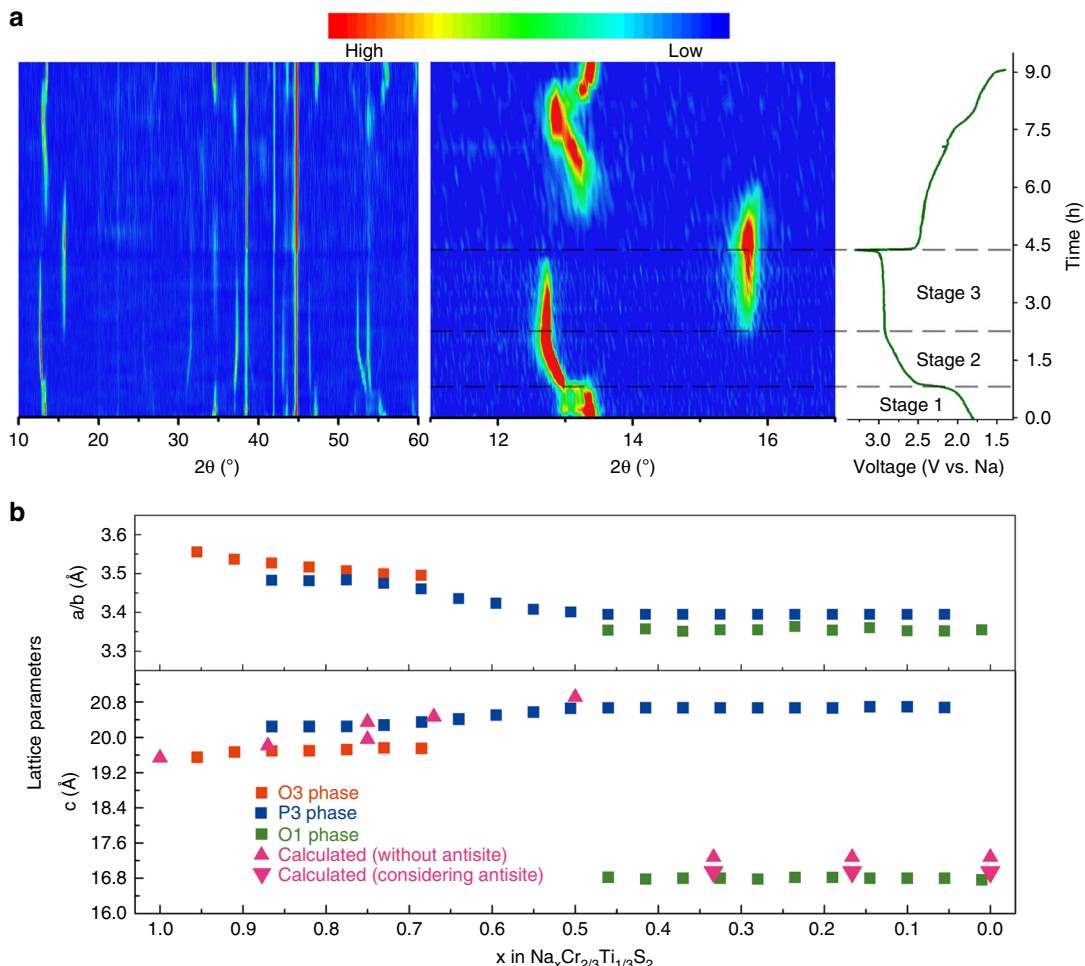

**Fig. 2** Structural evolution of $NaCr_{2/3}Ti_{1/3}S_2$ during cycling. **a** In situ XRD contour for $NaCr_{2/3}Ti_{1/3}S_2$ cell cycling at a rate of C/5. The constant intensity peaks at 38.54°, 41.98°, and 44.82° are peaks of Al foil. **b** Lattice parameters calculated from the in situ XRD and from density functional theory (DFT) calculations as functions of composition during the first charge. For easy comparison, c length of full charged O1′ phase is three timed to accord with O3/P3 phase

diffraction (SAED), and high-angle annular dark field-scanning transmission electron microscopy (HAADF-STEM). The Rietveld refined XRD schemes are presented in Fig. 4a, b. In the refined results, no crystalline impurity was observed. The structure parameters refined are listed in Supplementary Tables 3 and 4.

The microstructures are confirmed by SAED patterns and atomic scale HADDF-STEM images (Fig. 4c–f). The SAED pattern for pristine $NaCr_{2/3}Ti_{1/3}S_2$ and one-third charged $Na_{0.66}Cr_{2/3}Ti_{1/3}S_2$ (Fig. 4c, d) presents numerous diffraction points, and can be well indexed in the expected O3—the R-3m space group and P3—the P3m1 space group, respectively, which are in good agreement with the XRD results. In addition, we noticed that the shapes of diffraction points are regular, indicating good order of atoms without stacking faults or twin lamella. Figure 4e, f exhibit the images of HAADF-STEM for $NaCr_{2/3}Ti_{1/3}S_2$ and $Na_{0.66}Cr_{2/3}Ti_{1/3}S_2$ projected both along [100] direction at atomic-level, showing the typical layered structures. From the STEM images it is clear that the atoms are in good order in long range and can fit the predicted structures well, no disorder area or polymorphism was found. It can be certified from the images that during the early stages of first charging process there are no obvious signals in darker layers of Na, indicating that $Cr^{3+}$ ions were not migrated. Some stacking faults are captured in the following phase P3 $Na_{0.5}Cr_{2/3}Ti_{1/3}S_2$ (see

Supplementary Fig. 9), the corresponding SAED result showed a multiphase diffraction pattern. The slabs of S–Cr/Ti–S are obviously slid with Na ion kept in prismatic sites. Neither formation of S–S nor migration of Cr is observed in this phase, probably because the remaining Na can form Na–S ionic bond with S to stabilize the hole in S ion, similar to the reason that no O–O species formed in $Na_x[Mg_{0.28}Mn_{0.72}]O_2$[32], which has good cyclic performance as well. This could reveal the reason for great cyclic performance in stage 2.

To track the formation of S–S dimers and the migration of Cr ions in the full charged material, STEM images were used to analyze the microstructure. When observed along c-axis, apparent differences can be found between pristine O3 and full charged O1′ structures (Fig. 5a, b). In Fig. 5a, one spot represents an atom column consisting of Na, Cr, Ti, and S, showing a typical structure built by anionic close packing, with interior angles of polygons in regular 60° and 120°. In Fig. 5b, full charged O1′ structure shows a twisted polygon with interior angles in 67° and 113°. The brighter spots represent Cr/Ti cation columns and the darker spots represent S columns. We marked a hexagon built by S atoms and it can be seen that side lengths differ and there are two opposite sides, which are quite short. Each dotted triangle in the hexagon marked three S atoms in the same layer, the average length for sides of triangle are 0.293, 0.338, and 0.347 nm, respectively. In

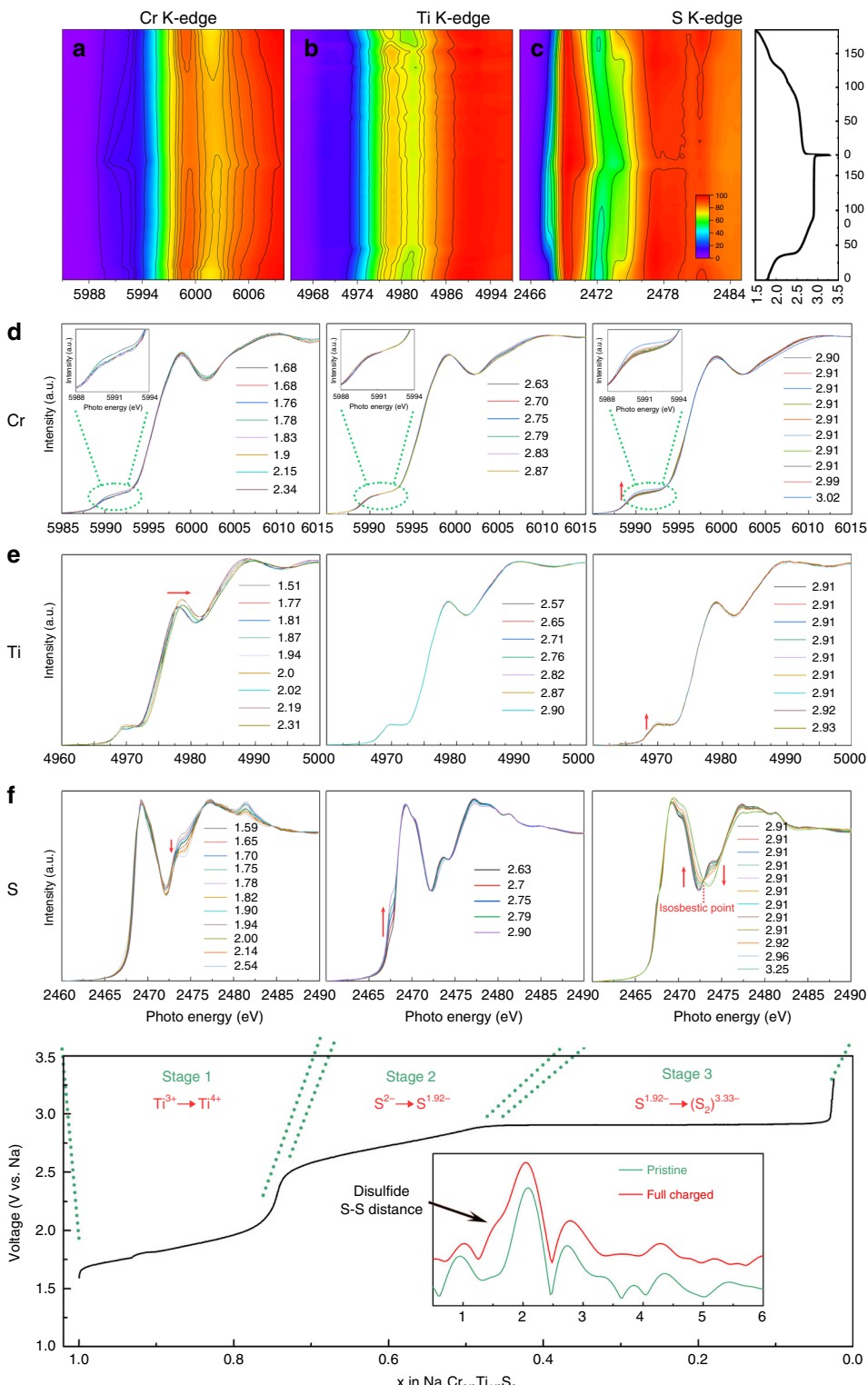

**Fig. 3** In situ K-edge X-ray absorption spectroscopy mapping. Mapping of **a** Cr, **b** Ti, and **c** S valance state, the corresponding voltage profiles for the first cycle is on the right side. **d**-**f** The divided in situ XAS K-edge XAS spectra of three stages as marked in the bottom inset during the first charge process. The inset in the bottom part is converted R space of ex situ sulfur K-edge EXAFS for pristine and full charged samples, showing an embedded peak of disulfide S–S distance

contrast, the S–S distance in the same layer in O3–NaCr$_{2/3}$Ti$_{1/3}$S$_2$ is 0.355 nm. That is a direct evidence for the formation of S–S dimers in the full charged structure, but that is not saying the S–S distance is exactly 0.293 nm, the formation of S–S dimer is caused by S atom moving around the equilibrium position, and the

STEM image is statistical average result. The S–S dimer could be formed between neighboring layers according to our calculation results discussed later, similar to Li's conclusion in Li-rich Li$_{1.2}$Ni$_{0.2}$Mn$_{0.6}$O$_2$[40]. In Fig. 5c and Supplementary Fig. 10 we observe the full charged sample along $a/b$-axis, from the layered

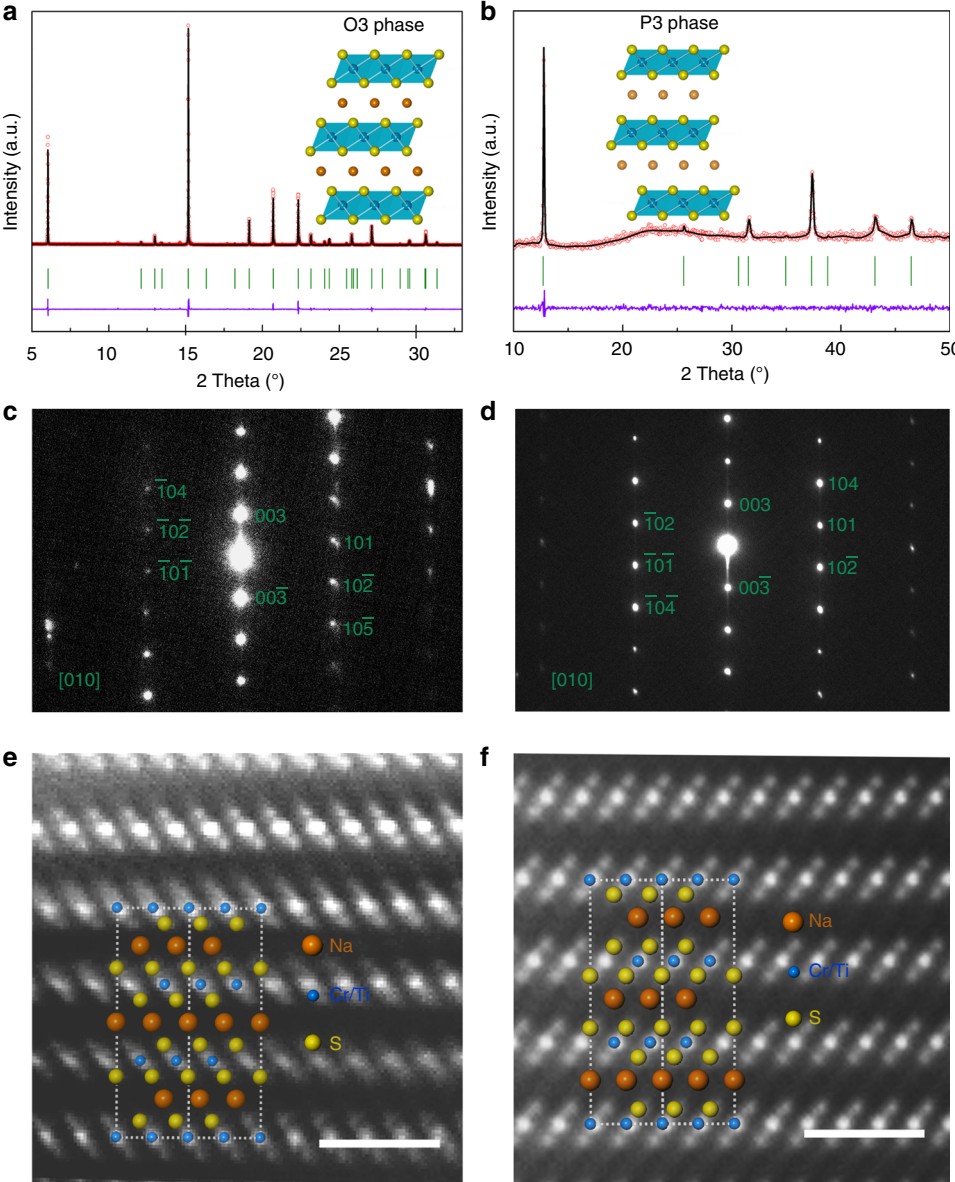

**Fig. 4** Structure characterization. **a** XRD refinement patterns of powder $NaCr_{2/3}Ti_{1/3}S_2$ (red circles), fitted profile (black solid line), and difference (purple solid line) using synchrotron XRD (wavelength 0.6884 Å). Bragg positions are indicated as green vertical tick marks. The inset is structure schematic of $NaCr_{2/3}Ti_{1/3}S_2$ observed along a-axis, legend: orange (Na), blue (Cr/Ti), and yellow balls (S). Impurity peaks at 11° and 14° can be attributed to $Na_2SO_3$ as a common impurity in commercial $Na_2S$. **b** XRD refinement patterns of powder $Na_{0.66}Cr_{2/3}Ti_{1/3}S_2$ and structure schematic of $Na_{0.66}Cr_{2/3}Ti_{1/3}S_2$ as inset. **c** Selected-area electron diffraction (SAED) pattern of $NaCr_{2/3}Ti_{1/3}S_2$ taken from [010] zone axis. **d** SAED pattern of $Na_{0.66}Cr_{2/3}Ti_{1/3}S_2$ taken from [010] zone axis. **e** HAADF-STEM image of the pristine $NaCr_{2/3}Ti_{1/3}S_2$ particle along a/b-axis, scale bar 1 nm. **f** HAADF-STEM image of the $Na_{0.66}Cr_{2/3}Ti_{1/3}S_2$ particle along a/b-axis, scale bar 1 nm

structure the Cr migration could clearly be seen. There are brighter points representing Cr atoms in the Na vacancies. From the intensity profiles between cation layers (on the right side) it can be seen that the blue colored peaks are much higher than the red colored peaks, so it can be concluded that blue peaks represents Cr/Ti atoms and red peak represent small amount of migrated Cr (the reason why Cr migrated instead of Ti will be discussed in calculation part by comparing the total energy and diffusion barrier), we analyzed the statistical average of peak areas and found the result between red/blue is ~0.24, that means ~29% of Cr migrated. The full deintercalation of Na can also be deduced from this figure for the uniform spacing between S layers accords with the typical spacing of S–Cr/Ti–S (0.27–0.29 nm for

$NaCr_{2/3}Ti_{1/3}S_2$ and $TiS_2$, respectively), much smaller than the S–Na–S spacing (0.37–0.44 nm for $NaCr_{2/3}Ti_{1/3}S_2$ and $Na_{0.5}Cr_{2/3}Ti_{1/3}S_2$). In addition, we selected three layers of S–Cr/Ti–S to show how the migration of Cr influences internal layer S–S distance and formation of S–S dimers (on the bottom side). Since the STEM image is average result of atom column, the peak intensity for Cr/Ti layer is much smoother than S layers, indicating irregular distribution of S around the equilibrium position. In situ EXAFS of Cr and ex situ EXAFS of Cr, Ti, and S were also performed to certify the migration as shown in Supplementary Figs. 11–13, respectively. In Supplementary Fig. 11a, the position of the first peak representing Cr–S distance decreases in the charging process and increases in the discharging

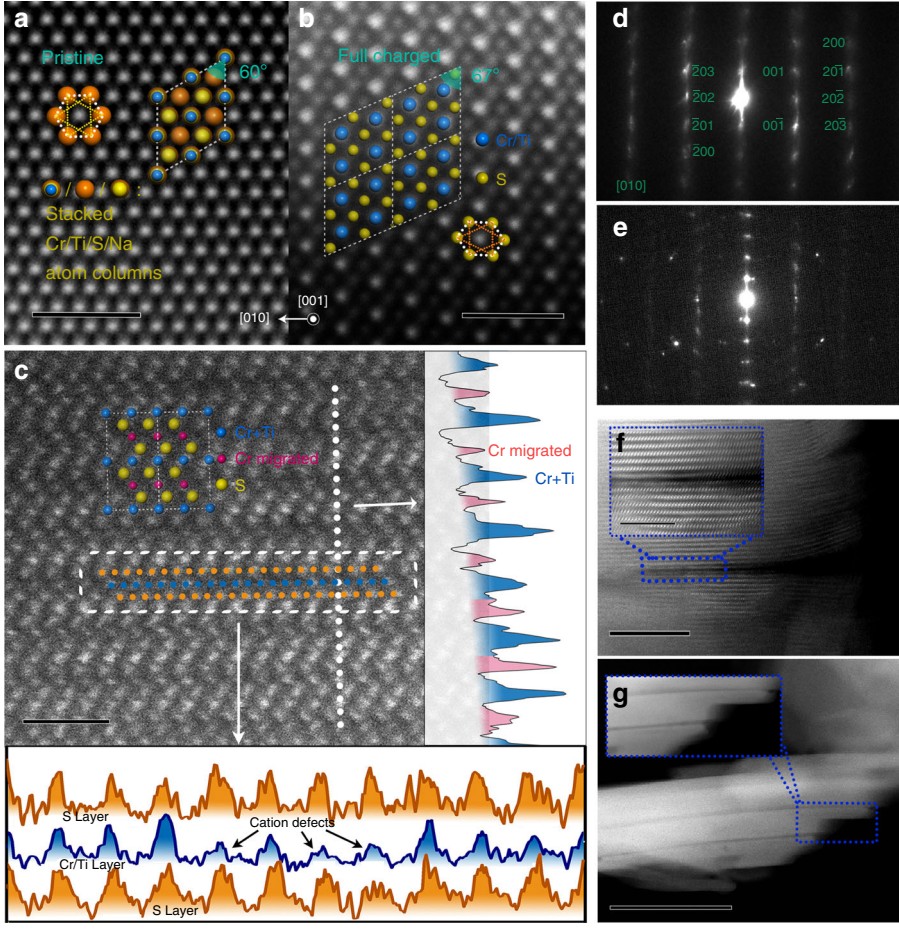

**Fig. 5** Characterization of full charged O1′ $NaCr_{2/3}Ti_{1/3}S_2$ and after cycling. STEM image of **a**, the pristine $NaCr_{2/3}Ti_{1/3}S_2$ particle along $c$-axis, scale bar 1 nm. **b** the full charged $Na_0Cr_{2/3}Ti_{1/3}S_2$ particle along $c$-axis, scale bar 1 nm. **c** The full charged $Na_0Cr_{2/3}Ti_{1/3}S_2$ particle along $a/b$-axis, scale bar 1 nm. **d** SAED pattern of full charged $Na_0Cr_{2/3}Ti_{1/3}S_2$ particle along [010]-axis. **e** SAED pattern of discharged sample after 10 cycles. **f** First one-third charged $Na_{0.66}Cr_{2/3}Ti_{1/3}S_2$ along $a/b$-axis, scale bar 10 nm, inset scale bar 5 nm. **g** Discharged sample after 10 cycles along $a/b$-axis, scale bar 100 nm

process, which is in good agreement with DFT calculation results (see Supplementary Fig. 14). The intensity of the second peak decreases in the charging process, indicating the generation of Cr vacancy and migration of Cr atoms. The second shell peaks do not completely disappear at charged state in in situ EXAFS data shown in Supplementary Fig. 10e, excluding the possibility of complete amorphization.

To confirm the existence of S–S bond, ex situ Raman test was performed. In Supplementary Fig. 15, the evolution of Raman peak can be observed at different charging states, a new peak at about 487 cm$^{-1}$ arose in the charging process and disappeared in discharging process. According to previous reports on $Li_2S_x$ ($x \geq$ 2) sulfides, the S–S dimers show a Raman shift in the range of 400–600 cm$^{-1}$[41]. And comparing to the samples like $FeS_2$, $CH–S–S–S–CH_3$ which are certified to have S–S bond, this Raman peak could be assigned for S–S ($\Delta v$). XPS measurements were also performed to confirm the valence of sulfur in the electrochemical process. In Supplementary Fig. 16 it can be seen that in pristine sample of $NaCr_{2/3}Ti_{1/3}S_2$ there are only peaks at 160.6 eV standing for $S^{2-}$ $2p3$[42] and after full charging the peaks of $S_2^{2-}$[43] and higher valence like elemental sulfur and $S_2O_3^{2-}$ appear. After discharging process, XPS peaks from elemental sulfur and $S_2O_3^{2-}$ are observed. These results verified the formation of $S_2^{2-}$, and that the elemental sulfur and its oxidized products are irreversible formed on the surface of electrodes.

In addition, we noted that the diffraction points (Fig. 5d) in O1′ image are more blurred if comparing to O3/P3 image in Fig. 4c, d, and there are diffraction lines along the $c$-axis, indicating anion defects cause by irregular distribution of S. The in situ XRD pattern of $Na_0Cr_{2/3}Ti_{1/3}S_2$ shows the same conclusion (Supplementary Fig. 17). (1 0–2) peak and (1 0 4) peak are broadened and their relative intensities to the (003) peak are lower than those values without any defects, indicating through-plane anion defects, similar to P2-type $Na_x[Fe_{1/2}Mn_{1/2}]O_2$[44]. These defects should be accompanied by the formation of S–S dimers.

In order to figure out the reason of capacity attenuation in Fig. 1b, the SAED pattern (Fig. 5e) of full discharged sample of $NaCr_{2/3}Ti_{1/3}S_2$ after 10 cycles was investigated. These diffraction lines along $c$-axis indicating S defect as well, and there is a set of impurity spots, stands for heterogeneous or amorphous phases, which can be directly seen in STEM image of the sample (Supplementary Fig. 18), these phases are likely to appear after several cycles and believed to be electrochemical inactive[18]. There is already a crack tip (Fig . 5f) in the first charging process. This kind of crack tip evolves into a long crack penetrating the whole crystal (Fig . 5g) after 10 cycles. Wang's group has systematically studied the reason for intragranular crack generation of $LiNi_{1/3}Mn_{1/3}Co_{1/3}O_2$ under high-voltage cycling, and found the deeper Li-ion extraction under high voltage and too large changes in

lattice are the primary causes[45]. In our study, there are indeed large changes in lattice during phase transition between $Na_{0.5}Cr_{2/3}Ti_{1/3}S_2$ and full charged $Na_0Cr_{2/3}Ti_{1/3}S_2$. The crack is parallel to the $c$-axis, corresponding to the change in $c$-axis length from 20.66 to 16.79 Å, which could be a main factor for crack generation. Near the edge of crack in the sample after 10 cycles, the intensity decay of sulfur layer is faster than that of Cr/Ti layer (Supplementary Fig. 19). The difference in intensity increased from 270 on 1.2 nm from the edge to 349 on the site of the edge, which means that the crack might appear accompanied by sulfur loss. The sulfur near the crack surface is tending to be oxidized by electrolyte or disproportionate into $S^{2-}$ and element sulfur, and the high valence oxidation products could be dissolved in electrolyte, which can further be confirmed by XPS measurements (Supplementary Fig. 20). We compared XPS spectra of the glass fiber separator before and after cycles, and found that the relative amount of S on glass fiber separator is increased by cycling. The peak area ratio between S 2$p$ and C 1$s$ is 0.04% for clean separator, 1.389% after 10 cycles and 3.445% after 50 cycles. It is clear that there is almost no sulfur in clean separator, and sulfur is precipitated in the form of elemental sulfur in the first 10 cycles, for its peak position is at 163.8 eV, accords with elemental sulfur[46]. But sulfur exists as higher valence on separator after 50 cycles, with peak position at 169.3 eV, we suppose this could be $S_2O_3{}^{2-}$ or polythionate because of the "tip effect"[47] on the surface of materials. Apparently, the formation of elemental sulfur should be responsible for the capacity attenuation and destruction of regular layer structure, resulting in the formation of localized heterogeneous or amorphous phases.

**Verifying the redox process theoretically**. To gain further insight into microstructure and electronic structure evolution of the charging process, density functional theory (DFT) calculations were performed to reveal the changes in crystal structure, total energy, bonding, density of state (DOS), and diffusion barriers. Figure 6a–c showed the DOSs of Cr, S, and Ti in $NaCr_{2/3}Ti_{1/3}S_2$, $Na_{0.58}Cr_{2/3}Ti_{1/3}S_2$, and $Na_0Cr_{2/3}Ti_{1/3}S_2$, respectively. In the pristine structure Cr/Ti are all about trivalent, Ti has electronic structure of $3d^1$, Cr has the electronic structure of $3d^3$ of high spin, with three electrons spinning up in $t_{2g}$ orbital. In previous reports[48] of $Na_xTiS_2$ and our experimental results of $NaTiS_2$ cathode (Supplementary Fig. 21), $Ti^{3+}$ cannot be totally oxidized to $Ti^{4+}$, leading to a relatively low capacity of ~160 mAh g$^{-1}$ comparing to theoretical capacity of 196 mAh g$^{-1}$, indicating that if charge transfer all comes from Ti, $Ti^{3+}$ will be oxidized only to $Ti^{3.8+}$ eventually. According to the electrochemical curves in Fig. 1b, oxidation to $Ti^{3.8+}$ corresponds to a capacity of 52 mAh g$^{-1}$, which is exactly corresponding to the inflection point of the curve at 2.1 V. It accords with the typical voltage range for the redox of $Ti^{3+}/Ti^{3.8+}$. And oxidation to $Ti^{4+}$ corresponds to a capacity of 64 mAh g$^{-1}$, which corresponds to the inflection point at 2.55 V. From the analysis above, we have figured out that the capacity comes from the oxidation of Ti and S. To figure out the degree of sulfur redox, the exact valence of Ti oxidation should be clear, we also analyzed the DOS of $Na_{0.75}Cr_{2/3}Ti_{1/3}S_2$ (Supplementary Fig. 22), when Ti is oxidized to $Ti^{3.75+}$, $Ti^{3.75+}$ still has occupied orbital at $E_f$, making it possible to be further oxidized to $Ti^{4+}$. The same conclusion of full oxidation of $Ti^{3+}$ to $Ti^{4+}$ can be drawn according to the research of Anton Van der Ven[49] in $NaTiS_2$ system, they found O1–P3 hybrid structure is stable at low Na concentration, which consists with our result. After the full oxidation of Ti, the orbital coupling of Cr 3$d$ and S 3$p$ can be seen from DOS image of $Na_{0.58}Cr_{2/3}Ti_{1/3}S_2$ in Fig. 6b, the DOS intensity of S 3$p$ near $E_f$ is higher than Cr 3$d$, indicating S is easier to lose electron and be

oxidized. These orbitals spread across $E_f$, which means the orbitals are underfilled, corresponding to the conclusion that sulfur provides electron and forms underfilled 3$p$ bands. In the full charged $Na_0Cr_{2/3}Ti_{1/3}S_2$, the DOS of Cr near $E_f$ is the highest and the energy band discontinuity from the splitting of $t_{2g}$ and $e_g$ is weaker, implying that the octahedral field effect is weakened due to the oxidation and deviation of sulfur. In Fig. 6d we show the calculated voltage platform comparing to experimental electrochemical curve. The calculated voltage matches with the experimental value, and we calculated the phase diagram (see Supplementary Fig. 23), it can be drawn that the structure evolution process calculated is in accord with the experimental results.

From the STEM images in Fig. 5c we confirmed the migration phenomenon of Cr, the migration of Cr makes it possible to form S–S bond in energy and orbital perspective. We calculated the total energy after different ratio of Cr migrated to Na vacancies. In Supplementary Fig. 24 it can be drawn that ~25% of Cr migration grants the lowest total energy, close to the 29% intensity ratio we drawn in STEM. From the optimized structure (Supplementary Fig. 25), we found the S–S bond or so-called S–S dimer formed between layers, and the S atom which formed S–S bond was once bonding with migrated Cr. From the perspective of orbital, the migration of Cr with Cr–S bonds broken will leave sulfur with isolated non-bonding $p$ orbital, which are capable of forming S–S bonds.

To calculate the diffusion barrier for Cr migration in $Na_0Cr_{2/3}Ti_{1/3}S_2$, climbing-image nudged elastic band (CINEB) method is used. As shown in Supplementary Fig. 26, $Cr^{3+}$ has lower diffusion barrier than $Ti^{4+}$ in the first step, 0.56 eV for Cr and 1.69 eV for Ti, respectively, mainly for $Ti^{4+}$ has higher electrostatic force towards S anion. The diffusion path is shown in Supplementary Fig. 27. The diffusion barrier of Cr migration for the second is 0.83 V, but the total energy after 2 Cr atoms (25% Cr in $2 \times 2 \times 1$ cell) migrated is 1.79 eV lower than the original structure, which could provide enough energy. The reduction of energy after Cr migration mainly comes from the bond energy released in the formation of S–S and the weakening of Van der Waals repulsion between S layers.

**Discussion**
The formation of S–S dimers is directly observed in a model layered $Na_xCr_{2/3}Ti_{1/3}S_2$ and the migration of Cr to Na vacancies is revealed. In previous works on O3 $LiMO_2$ materials[50], transition metal migration is considered to be hardly reversible, and exerts negative effects, such as capacity loss, lower ion diffusivity, and increased electronic resistance. The migration of transition metal ions may lead to complete conversion to spinel type structure upon de-intercalating the alkali metal ions. However, our results claimed that the migration of Cr in $NaCr_{1-y}Ti_yS_2$ is reversible from P3 to O1′. Instead, the loss of sulfur is the key factor responsible for the capacity loss. Based on in situ XRD, in situ XAS, ex situ STEM, SAED, Raman, XPS, and DFT results, various anionic redox chemistries are proposed,

$$S^{2-} - (2-n)e^- \leftrightarrow S^{n-} (\textbf{Electron holes})(1.92 < n < 2) \quad (1)$$

$$2S^{1.92-} - (3.84-n)e^- \leftrightarrow (S_2)^{n-} (\textbf{Dimers})(3.33 < n < 3.84) \quad (2)$$

$$2S^{2-} - 2e^- \leftrightarrow (S_2)^{2-} (\textbf{Disulfides}) \quad (3)$$

$$(S_2)^{2-} - 2e^- \rightarrow (S_2)(\textbf{Sulfur}) \quad (4)$$

We define S–S dimer as $(S_2)^{n-}$ (broadly speaking, $2 \leq n < 4$) here, the disulfides are defined as exactly $(S_2)^{2-}$, which can be

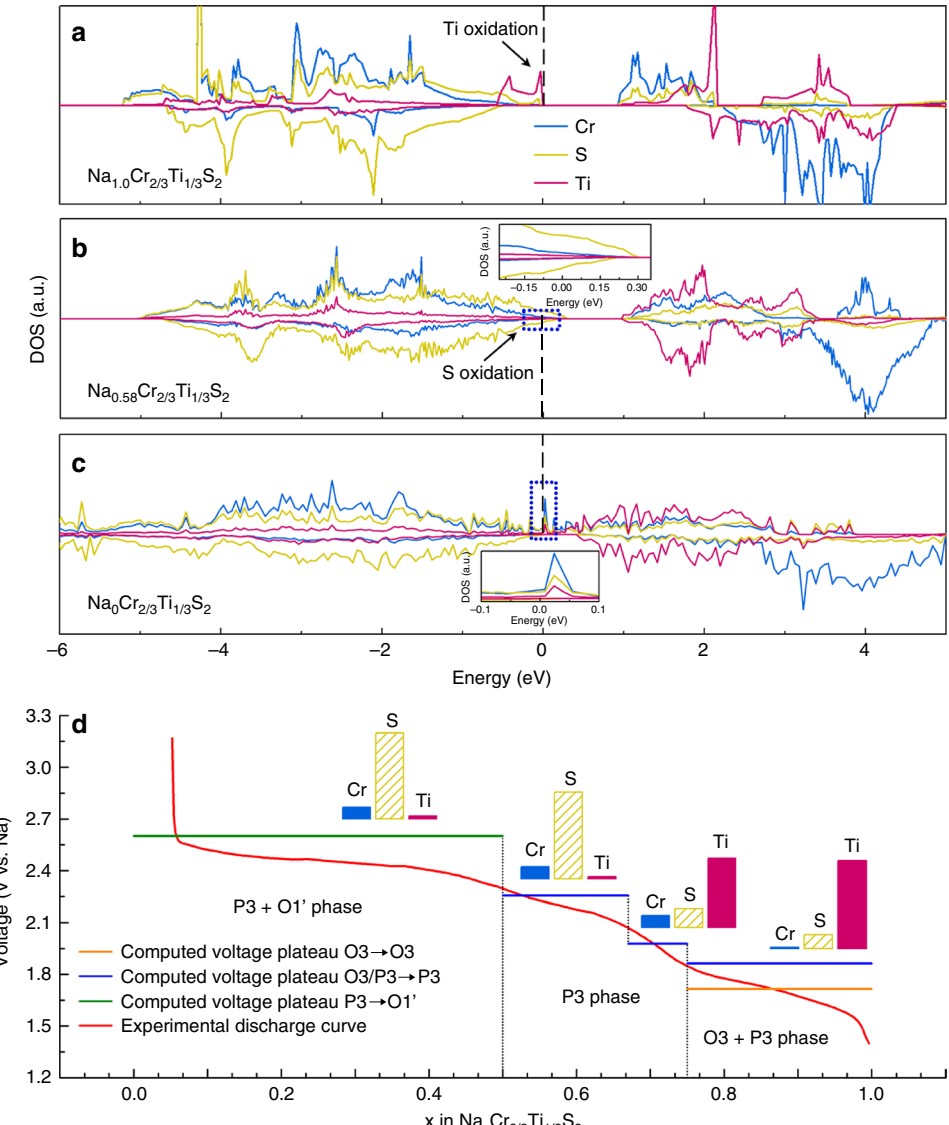

**Fig. 6** Density of states and computed voltages. The DOS of **a** $NaCr_{2/3}Ti_{1/3}S_2$, **b** $Na_{0.58}Cr_{2/3}Ti_{1/3}S_2$, and **c** $Na_0Cr_{2/3}Ti_{1/3}S_2$ with the Fermi level marked using vertical dashed lines. **d** Calculated voltage platform comparing to experimental electrochemical curve. The columns above the platform are contribution of elements in the redox process, coming from integration of DOS in the range of −0.1 to 0 eV

considered as a special case of S–S dimer. The localized electron holes ($S^{n-}$) are formed at the top of a given $3p$ band with Na ions removed at the P3 evolution process in order to compensate the charge balance as expressed in Eq. (1). The crystal-field stabilization of the sulfur sublattice is degraded upon the further removal of $Na^+$ ions. Consequently, the bonding of an oxidized $S^{n-}$ ion to its original lattice site is weakened, which leads to the local stacking faults of sulfur sublattice at the end of this process. When more than 0.5 Na ions are extracted, the part-misplacement of layer sulfur in the sublattice, along with the weakening of Cr–S electrostatic attraction due to the decrease of the charge on sulfur induce ~1/4 Cr ions to migrate to Na vacancy sites, resulting in the structural transformations from P3 to O1′. The formation of $(S_2)^{n-}$ (Eq. (2)) dimers in O1′ phase is observed by STEM, which directly shows the shortening of the S–S distance to around 0.293 nm. In the meantime, the anion defects and the formation of $(S_2)^{2-}$ (Eq. (3)) in O1′ structure are accompanied during the Cr migration process. The irreversible oxidation or disproportionation of $S_2^{2-}$ pairs to sulfur occurs

near the crack surface as mentioned before (Eq. (4)). The structural flexibility of the sulfur network can be assumed, as well as repeated cationic migrations and the breathing of S–S dimers upon cycle process. In our case, $S^{2-}/S^{n-}$ (Eq. (1)) and $S^{n-}/(S_2)^{n-}$ (Eq. (2)) redox processes are similar to two types of oxygen redox in $Li_xM_yO_z$ that were reported previously, e.g., $O^{n-}$ in $Li_{1.2}Ni_{0.13}Co_{0.13}Mn_{0.54}O_2$[17] and $(O_2)^{n-}$ in $Li_2IrO_3$[11]. It can be concluded that localized electron holes on the anions and isolated non-bonding anion $p$ orbitals are prerequisites to form $(S_2)^{n-}$ species, which is in accord with the mechanism elucidated in oxide electrode systems[17].

In this work, a novel layered $NaCr_{2/3}Ti_{1/3}S_2$ is successfully synthesized and the capacity is boosted from ~100 mAh g$^{-1}$ in $NaCrS_2$ to ~186 mAh g$^{-1}$ in this system. The extra capacity is attributed to a synergy effect on the anionic (($S^{2-}/S^{n-}$), $(S^{n-}/(S_2)^{n-})$) and cationic ($Ti^{3+/4+}$) redox. The electrochemical effect of $Ti^{3+}$ dopants, which has $3d$ orbital above S $3p$ orbital, can provide ~1/3 of the capacity. In addition, it makes the structure transform to a more stable P3 phase in low Na content

and the oxidized $Ti^{4+}$ can stabilize the structure as well. The dimers in $O1'$ are stabilized by $Cr^{3+}/Ti^{4+} - (S_2)^{n-}$ covalent interactions as the bottom portion of S-$3p$ band merges with the Fermi level. Various anionic redoxes of sulfur are controllable at different state of charge. Normally, it is difficult to distinguish different kinds of anionic redox activities because of the composition and structural complexity of the electrode. Our experimental results provide core evidence to reveal various charge compensation mechanisms, involving the formation of electron holes, anionic dimers, disulfide-like species, as well as the precipitation of elemental sulfur. Our results have demonstrated possible routes of anionic redox processes from electron holes to element. These results provide a insight to clarify the nature of an anionic redox process and help to establish a comprehensive scientific understanding for anionic chemistry.

## Methods

**Sample preparation**. To synthesize the $NaCr_{1-y}Ti_yS_2$ series materials, the mixture of $Na_2S$, Ti, S, and Cr powder in a stoichiometry of 3:2:9:4 was compressed and placed into carbon-coated quartz tubes. Then the mixture was slowly heated to 700 °C, kept for 10 h, and then cooled down slowly for over 5 h to room temperature. All preparation was under argon atmosphere unless otherwise noted.

**Electrochemical characterization**. A slurry of as-prepared $NaCr_{1-y}Ti_yS_2$ (70 wt %), conductive carbon black (20 wt%), and polyvinylidenefluoride (PVDF, Sigma-Aldrich, 10 wt%) dispersed in N-methyl-2-pyrrolidone (NMP, Sigma-Aldrich) was coated on aluminum foil. The electrolyte consisted of 1 M $NaClO_4$ dissolved in 19:19:2 volume ethylene carbonate/dimethyl carbonate/fluoroethylene carbonate. Then 2032-type coin cell was assembled for electrochemical test. Electrochemical measurements were tested on LAND battery testing system.

**XRD measurements**. Synchrotron XRD data were recorded at Shanghai Synchrotron Sadiation Facility (SSRF, beamline BL14B1). The wavelength of the X-ray was 0.6884 Å. The angles were converted to the angles of Cu-Ka ($\lambda = 1.54$ Å), in order to be consistent with other data, then refined using the GSAS software based on the Rietveld method. The in situ XRD were tested at the Institute of Physics, Chinese Academy of Sciences, using X'Pert Pro MPD X-ray diffractometer (D8 Bruker) equipped with Cu-Ka radiation ($\lambda = 1.5405$ Å) in the scan range ($2\theta$) of 10–60° under a current density of 0.2 C to avoid deterioration, using a cell with small windows sealed with Al foil.

**XAS measurements**. In situ Cr/Ti/S K-edge XAS spectra were measured at Taiwan Light Source (Beamline 16A1) at National Synchrotron Radiation Research Center. The bending magnet beamline delivers monochromatic photon beams with energies from 2 to 8 keV with a resolving power ($E/\Delta E$) up to 7000, and a beam size of 0.5 mm ($H$) × 0.4 mm ($V$). The XAS were collected using total fluorescence yield method with the sample chamber filled with helium gas[51]. The XANES spectra were processed using Athena software packages[52]. 2D XRF images and ex situ S K-edge XAS spectra of the pristine and charged sample were collected at beamline 8-BM (TES) of the National Synchrotron Light Source II (NSLSII) at Brookhaven National Laboratory (BNL). 2*2 mm² area from the middle of the the electrodes were selected for the XRF measurement. XRF images were collected in a continuous scan mode with a 20 μm of pixel size at a energy of 3000 eV, which is above the absorption edge of S and Cl. Ex situ XAS spectra at Ti and Cr K-edge were also collected at beamline 7-BM of the National Synchrotron Light Sources II (NSLS-II) at Brookhaven National Laboratory (BNL).

**SEM measurements**. The measurements were obtained using a field emission scanning electron microscope (Cambridge S-360) equipped with an energy-dispersive X-ray spectroscopy (EDS) detector for elements analysis. The images were obtained by 12 kV voltage.

**STEM measurements**. The measurements were obtained by a field-emission transmission electron microscope (JEOL ARM200F, 200 keV) with spherical aberration corrector for the probe-forming lenses, which was operated at 200 kV. The Cs corrector was optimized for STEM observations and the point-to-point resolution in the STEM mode is better than 1.0 Å. For HAADF-STEM imaging, the electron probe convergence angle of 25 mrad and an HAADF detector with an inner angle larger than 100 mrad were used. The HAADF detector was set to collect electrons scattered between 100 and 267 mrad, guaranteeing that the collected signals give an approximately incoherent atomic-number contrast in HAADF-STEM.

**XPS characterization**. XPS was carried out on a PHI 5000C ESCA System with monochromatic Al-Ka X-ray source. The C $1s$ peak at 285.0 eV from hydrocarbon contamination was used to calibrate the binding energy.

**DFT + U calculations**. Based on the projector-augmented wave (PAW) method within DFT theory[53], conducted with the VASP program[54], spin-polarized calculations were carried out. We used the Perdew–Burke–Ernzerhof functional for exchange correlation[55]. We set an effective $U_{eff}$ value to 3.5 eV for Cr and 3.2 eV for Ti as discussed in electronic structure calculations on $MCrS_2$ (M=Li, Na, K, and Ag)[56,57]. The plane wave cutoff energy and Monkhorst–Pack $k$-point mesh were set to 550 eV and $3 \times 3 \times 3$ for $Na_xCr_{2/3}Ti_{1/3}S_2$ $2 \times 2 \times 1$ supercells. As for the calculation of the electronic DOSs, $5 \times 5 \times 3$ $k$-point mesh for the conventional cell and the modified tetrahedron method were used. The above parameters made the total energy converged to 2 meV per atom. The calculated structural parameters of $Na_xCr_{2/3}Ti_{1/3}S_2$ are consistent with experimental ones as shown in Fig. 2a. To determine the energy barriers for Na or Cr ion diffusion in $Na_0Cr_{2/3}Ti_{1/3}S_2$ the CINEB method[58] was employed for searching the minimum-energy path.

## Data availability

Data supporting this study are available in the article and corresponding Supplementary Information files. Extra data or the source data are available from the corresponding authors.

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

## Acknowledgements

This work was financially supported by the National Natural Science Foundation of China (Grant No. 21773037, 21473235, 11227902, 11704245 and U1632269); the National Key Scientific Research Project (Grant No. 2016YFB090150); Shanghai Pujiang Program (17PJ1403700). The work at Taiwan Light Source (beamline 16A1) was supported by National Synchrotron Radiation Research Center. The work at Brookhaven National Laboratory was supported by the Assistant Secretary for Energy Efficiency and Renewable Energy, Vehicle Technology Office of the U.S. Department of Energy through the Advanced Battery Materials Research (BMR) Program, including Battery500 Consortium under contract DE-SC0012704. This research used resources at beamlines 7-BM (QAS), 8-BM (TES) of the National Synchrotron Light Source II, a U.S. Department of Energy (DOE) Office of Science User Facility operated for the DOE Office of Science by Brookhaven National Laboratory under Contract no. DE-SC0012704. DFT calculation was performed in National Supercomputing Center in Shenzhen (Shenzhen Cloud Computing Center). P.L. is supported by the Program for Professor of Special Appointment (Eastern Scholar) at Shanghai Institutions of Higher Learning. The authors also thank beamline BL14B1 of the Shanghai Synchrotron Radiation Facility (SSRF). G.-X.R. also acknowledges the support from the UCAS Joint Ph.D. Training Program. The authors gratefully acknowledge the help by Prof. Xi-Qian Yu for his help on in situ XRD measurements, Bo-Yuan Ning, and Hong-Kun Zhu at Fudan University for their help on DFT calculations, Peng-Fei Yu, Xin-Yang Yue, Qin-Chao Wang, Ding-Ren Shi, Jing-Ke Meng, Wei-Wen Wang, Si-Yu Yang and He-Yi Xia for their help on experiments.

## Author contributions

Z.-W.F. and X.-S.L. supervised the whole project. Z.-W.F. and Z.S. designed the experiment. T.W., X.-Q.Y., X.-S.L., and Z.-W.F. wrote the manuscript. T.W. and M.-H.C. tested the electrochemical performance. J.-N.Z. performed the in situ XRD measurements. T.W. and J.-L.Y. carried out the DFT calculations. G.-X.R. and X.-S.L. performed the K-edge XANES experiments as well as processed the data. Z.S., S.-M.B., X.-Q.Y. and P.N. measured the 2D XRF images and ex situ S K-edge XAS data. P.L. and M.-W.C. conducted STEM imaging.

## Competing interests

The authors declare no competing interests.
