## [Peer Review File · Nature Communications]

Review of « Anionic redox reaction in layered $\text{NaCr}_{2/3}\text{Ti}_{1/3}\text{S}_2$ through electron holes formation in addition to the dimerization of S-S”

The authors studied an O3- $\text{NaCr}_{2/3}\text{Ti}_{1/3}\text{S}_2$ cathode, delivering a capacity beyond the $\text{Ti}^{3+}/\text{Ti}^{4+}$ redox limit based on combined cationic and anionic redox processes. Various charge compensation mechanisms involving sulfur have been studied/discussed such as formation of localized holes ($\text{S}^{\cdot-}$), S_2^{2-} species, S-S dimers ($\text{S}_2^{\cdot-}$), and dissolution of elemental Sulfur in the electrolyte. The authors used a wide range of relevant characterization techniques to characterize the cationic (in situ Ti/Cr K-edge XANES) and anionic redox (in situ S K-edge XANES, Raman, XPS...) and cation migration (STEM). However, the crystallographic study needs to be revised and the presence (absence) of migrating Cr (Ti) and SS dimers formation need to be confirm by analysis of the EXAFS part of the S, Cr, Ti K-edges XAS data. Therefore, this paper should be accepted after the applying/answering the following revisions/comments:

- Synthesis:
 - The authors said that the structure of all $\text{NaCr}_{1-y}\text{Ti}_y\text{S}_2$ composition are similar to NaTiS_2 and NaCrS_2 but the diffraction pattern of NaTiS_2 looks quite different to the others.
- Electrochemical tests:
 - The Columbic efficiencies at the first cycle are superior to 100% (for $y=1, 0.5$ and 0.33). Is it really possible to insert more than 1 Na^+ in $\text{Na}_x\text{Cr}_{1-y}\text{Ti}_y\text{S}_2$ materials? Na vacancies at the pristine state, often reported for the synthetize NaTMO_2 without Na precursor excess, should be the reason.
- in situ XAS:
 - the author mentioned that *Cr Pre-edge intensity increases “in stage 3 that might be resulting from the local distortion of Cr-S octahedron caused by movement of S”* but an increase of Cr pre-edge is also observed during stage 1 where no distortion of S network is expected (*i.e.* no formation of SS dimers). Could you comment?
 - *“The gradually increasing intensity of pre-edge at 4970 eV indicates the oxidation process from Ti^{3+} to Ti^{4+} in the first stage of charging process”*. The oxidation of Ti^{3+} to Ti^{4+} isn't supposed to lead to an increase of the pre-edge intensity as Ti^{3+} (t_{2g}^1) is Jahn Teller active whereas Ti^{4+} (t_{2g}^0) isn't. Could you comment?
 - *“The pre-edge of Ti exhibited the same phenomenon as the spectra of Cr due to the comparable chemical environment of Cr-S and Ti-S”*. Looking the Ti K-edge data carefully, I can't see any increase of Ti K pre-edge during stage 3.
 - *“First of all, during the first stage, the S K-edge is hardly changed”*. The authors have assigned the first stage to $\text{Ti}^{3+}/\text{Ti}^{4+}$ redox. If cationic redox involve in this first stage, the pre-edge intensity should increase through the generation of holes in the

admixed $\text{TM}_{3d}\text{S}_{3p}$. Moreover, the authors said: “*In the second stage, the pre-edge located at ~ 2468 eV is gradually grown*”. This phenomenon is usually attributed to cationic redox via holes formation in the admixed $\text{TM}_{3d}\text{S}_{3p}$. Could you comment?

- “*the distorting of CrS_6 octahedral structure upon the removal of Na ion should be responsible for the formation of the typical pre-edge*” The pre-edge intensity can be assigned to a distortion for TM K-edge because for those elements the pre-edge corresponds to the $1s \rightarrow 3d$ transition which is a forbidden (due to symmetry arguments) and become allowed due to the distortion, breaking the symmetry. Whereas for $1s \rightarrow 3p$ (allowed transition) the distortion isn’t supposed to have any significant effect on the pre-edge intensity.
- According to your S K-edge XANES data, you conclude to the existence of S_2^{n-} but this hypothesis **should be checked with S K-edge EXAFS** part of XAS data collected. Such nice XANES (*i.e.* not noisy) data should give good enough EXAFS oscillations allowing to detect any formation of S-S dimers or S_2^{2-} species. **Moreover Cr/Ti K-edge EXAFS should give an experimental proof of their possible migration in empty Na site.**
- Rietveld refinements:
 - The authors should mention that there are lines at around 11° and 14° which aren’t fitted with this model and could corresponds to impurity lines or superstructure peaks associated with an eventual Cr/Ti ordering.
 - I can’t imagine that the $\text{Na}(\text{Cr}_{2/3}\text{Ti}_{1/3})\text{S}_2$ and $\text{Na}_{0.67}(\text{Cr}_{2/3}\text{Ti}_{1/3})\text{S}_2$ phases crystallize in a P1 space group as mentioned in table S2 and S3. The cell parameters are very close a hexagonal cell which is the classical space group for the O3 and P3 structures. They can sometime be described in monoclinic symmetry when honeycomb-like superstructure peaks appear or orthorhombic cell when a Jahn Teller distortion occurs but never in triclinic cell. **The author should perform the Rietveld refinement using the appropriated space group.**
 - χ^2 can’t be < 1 , it means that the model fits the noise due to the refinement of too many parameters compared to the number of experimental information available (using P1 instead of R-3m space group should be the reason of such low χ^2 value).
 - The Wyckoff position are also wrong, using P1 space group, there is no symmetry elements so only 1 Wyckoff position possible: the 1a.
 - It’s not possible to refine Ti/Cr partial occupancies based on XRD data. There are only 2 electrons difference, too low to be refined: **partial occupancies need be fixed according to the composition suggested by chemical analysis.**

- STEM:
 - The utilization of STEM to identify some Cr in the Na layer could be inconclusive. There are obvious stacking faults (occurring along the 001 direction but also along other directions, as evidenced by the obvious anisotropic broadening of the diffraction lines seen on the XRD pattern of the full charged sample) mitigating the contrast between TM layer and “empty” Na layer. Based on STEM, the authors estimate that 25-30% of Cr migrate in empty Na site, **such huge amount of anti-site defects should be seen thanks to Cr K-edge EXAFS. Moreover Ti K-edge EXAFS should also give the experimental proof the absence of Ti migration that the authors claim.**
 - The authors claim that the formation of the SS bond is associated to the Cr migration in its neighboring. But, according to the Fig 5, the SS distances don't seem to depend on the amount of migrating Cr. Moreover the “short SS distance” (0.2 nm) that authors claimed to detect looks like noise: it's probably an over-interpretation of the data.
- Raman:
 - The corresponding capacity should be mentioned on the ex-situ Raman figure (it's difficult to compare with the electrochemistry)
 - The peak assigned to S-S dimers in $\text{Na}_x(\text{Cr,Ti})\text{S}_2$ looks higher for sample#3 than #4 and #5 (end of charge), whereas according to your theory the SS dimers are supposed to be formed only from the $x \sim 0.45$, approximately the composition of the sample#3.
 - FeS_2 is supposed to have SS bonds but doesn't show any contribution in the “SS dimers” region. Could you comment?
- General comment: Please be careful about the terms employed and the references chosen
 - Line 57-58: speaking about Li-Rich NMC: “Oxygen anion redox with the reversible formation and decomposition of O–O dimers is witnessed^{17,20,21}.” Those references refer to Li_2IrO_3 , Li_2RuO_3 and Na_3RuO_4 and not to Li-Rich NMC.
 - Line 198/199: “O-O species formed in $\text{Na}_x[\text{Mg}_{0.28}\text{Mn}_{0.72}]\text{O}_2$ ²⁵”. The authors of this paper never mentioned any formation of O-O species in this compound.
 - Lines 387/389 : “In our case, $\text{S}^{2-}/\text{S}^{n-}$ (eq.1) and $\text{S}^{n-}/(\text{S}_2)^{n-}$ (eq.2) redox processes are similar to two types of oxygen redoxes in $\text{Li}_x\text{M}_y\text{O}_z$ reported previously, e.g. O^{n-} in $\text{Li}_{1.2}\text{Ni}_{0.13}\text{Co}_{0.13}\text{Mn}_{0.54}\text{O}_2$ and $(\text{O}_2)^{n-}$ in Li_2IrO_3 ”. That's true but it's in disagreement with the comment made at Line 57/58

In conclusion this article should be accepted after applying the revision proposed especially regarding the Rietveld refinements and treatment of the EXAFS part of the S/Cr/Ti K-edges XAS. Moreover the authors shouldn't over-interpret the data (*i.e.* some STEM data) and need to clarify discrepancies (*i.e.* SS dimers formation seen by ex-situ Raman vs. state of charge).

Reviewer #2 (Remarks to the Author):

The authors investigated anionic redox in layered $\text{NaCr}_{2/3}\text{Ti}_{1/3}\text{S}_2$. Structural characterization was carried out to elucidate S-S formation and the mechanism of anionic redox for the active material. There was a suite of characterization techniques including XRD, XAS, HAADF-STEM, Raman, XPS, and electronic structure calculations. While this study employs many techniques, the reviewer is not convinced of dimerized S-S formation from sulfide species in the pristine material. Below are questions for clarifying some of the data supporting dimerization of S-S.

Anionic dimer formation was supported by X-ray absorption spectroscopy. However, the plots are difficult to digest. While Stage 1 in Figure 3 shows no sulfide oxidation state change as evidenced by the lack of change in the S K-edge, Stage 3 depicts quite a bit of change in the post-edge region but not much in the pre-edge region. Is the sulfur dimerization occurring without changes in the oxidation state? The XAS data seems to support structural rearrangement. and not the $\text{S}(-1.92 \text{ charge})$ to S_2 with 3.33 charge per disulfide.

The XPS data to suggest formation of oxidized forms of S (e.g. sulfide and sulfites) is depicted in supplementary Figure 9 (bottom pane). However, the fitting of pristine sulfide (S^{2-}) in $\text{NaCr}_{2/3}\text{Ti}_{1/3}\text{S}_2$ is quite poor. The $2p_{3/2}$ and $2p_{1/2}$ peaks are not well resolved for S 2p binding energies. Also, there is consistent notation that is awkward and unconventional (e.g. $2p_3$ vs. $2p_{3/2}$).

The S-S formation was also reasoned to occur as indicated by Raman peak at 487 cm^{-1} . However, as has been demonstrated with Li-S (from S catholyte) and Na-S reports, polysulfides will form and possess normal modes over a range of $400\text{-}600 \text{ cm}^{-1}$.

What is the mechanism of disproportionation of sulfide species near the crack in the electrode to produce both S^{2-} species (also, what are these species) and elemental sulfur when desodiating from $\text{Na}=0.5$ to $\text{Na}=0$ in $\text{Na}(x)\text{Cr}_{2/3}\text{Ti}_{1/3}\text{S}_2$?

Reviewer #3 (Remarks to the Author):

This paper reports anionic redox processes in sulfide electrode $\text{NaCr}_{2/3}\text{Ti}_{1/3}\text{S}_2$ for Na ion batteries and their extensive characterization with most modern and sophisticated physical-chemical methods, with emphasis on structural characterization. This is a hot topic, with many papers being published on sodium ion batteries every year. The authors have done a substantial amount of work on synthesis and characterization of their model electrode material, accumulated extensive data, analyzed them carefully and observed some interesting trends revealing the importance of anionic processes. I think that the paper is suitable for publication after minor revision. My comments are listed below:

- 1) Do you present electrochemical measurements for only one coin cell? Have you tried to assemble several ones, and if so, how reproducible were the results?
- 2) In the abstract, you mention that the capacity is 190 mAh/g, but this number is only achieved in the first cycle. For Li-ion batteries, the 1st discharge-charge cycle is often referred to as a "formation cycle", shouldn't you put in the Abstract the capacity value, which is more accurately reflects the real performance?
Or, the data in Figure 1 represents actually the data for capacity retention after 50 cycles based on the second cycle discharge capacity, that is, after the formation cycle?
- 3) In the Abstract, the expression "anion redox reaction between S_2^- and $(\text{S}_2)^{2-}$ " is confusing, as it sounds like these two groups interact with each other, while in fact they transform to each other.

The sentence "In this study, various charge compensation mechanisms of sulfur anionic redox process

in layered $\text{NaCr}_2/3\text{Ti}_1/3\text{S}_2$ through the formation of electron holes, disulfide-like species, the precipitation of elemental sulfur, as well as S-S dimerization, especially through the formation of electron holes are investigated" seems to be somewhat redundant.

"formation of ... disulfide-like species" and "S-S dimerization" are the same?

"formation of electron holes"... "especially through the formation of electron holes"?.. mentioned twice

4) As you discuss the advantages of the anionic redox processes, it would be helpful to comment, how your mixed-metal material compares to other polysulfide electrodes operating due to redox in disulfide group $(\text{S-S})_2^- + 2e^- = 2\text{S}_2^-$ (for example, TiS_3).

As Nature communications is a multidisciplinary journal, for a wider audience, it would be helpful to put your work in a more general context.

5) Figure 5 c for the fully charged sample implies that the Cr migration phenomenon is rather pronounced (29 % (25 % by DFT) as stated in the paper), so, do you have a correspondingly significant number of S atoms paired in dimers in the fully charged sample? So, not only S atoms at the edges of the layers form dimers?

Fig. S18 is not completely clear.

The terms "dimers" and "disulfides" seem to be used in different meanings, which is a bit confusing and should be defined.

6) Please specify more clearly, whether the 4 equations in the Discussion represent 4 consecutive steps? If so, than in the Abstract the phrases like "other anion redox mechanism" are inaccurate, as this is not another mechanism, but just a step in a more complex process.

7) How did you obtain the numbers 3.33 and 3.84 for charge in the Eq. (2)?

8) Recent reviews on anionic redox in electrode materials should be cited:

Grayfer, E. D. et al. (2017). Anionic redox chemistry in polysulfide electrode materials for rechargeable batteries. *ChemSusChem*, 10(24), 4805-4811.

Zhao, C. et al. (2017). Review on anionic redox for high-capacity lithium-and sodium-ion batteries. *Journal of Physics D: Applied Physics*, 50(18), 183001.

Yang, W., & Devereaux, T. P. (2018). Anionic and cationic redox and interfaces in batteries: Advances from soft X-ray absorption spectroscopy to resonant inelastic scattering. *Journal of Power Sources*, 389, 188-197.

9) At some (not numerous) points, grammar/typos should be revised, like "possible routes of anionic redox processes from the electron holes to element is" – routes are

"The localized electron holes (Sn^-) is formed" are formed

"the capacity from ~ 100 mAh/g in NaCrS_2 to ~ 190 mAh/g." sounds like incomplete sentence

"Sample preparation. To synthesize the $\text{NaCr}_{1-y}\text{Ti}_y\text{S}_2$ series materials" – series?

"the mixture of Na_2S , Ti, S, and Cr powder in stoichiometry was compressed" - which stoichiometry?

The changes and responses to the reviewers are listed below point by point:

Reviewer #1 (Remarks to the Author):

Review of « Anionic redox reaction in layered $\text{NaCr}_{2/3}\text{Ti}_{1/3}\text{S}_2$ through electron holes formation in addition to the dimerization of S-S»

Comments:

The authors studied an O3- $\text{NaCr}_{2/3}\text{Ti}_{1/3}\text{S}_2$ cathode, delivering a capacity beyond the $\text{Ti}^{3+}/\text{Ti}^{4+}$ redox limit based on combined cationic and anionic redox processes. Various charge compensation mechanisms involving sulfur have been studied/discussed such as formation of localized holes ($\text{S}^{\cdot-}$), S_2^{2-} -species, S-S dimers ($\text{S}_2^{\cdot-}$), and dissolution of elemental Sulfur in the electrolyte. The authors used a wide range of relevant characterization techniques to characterize the cationic (in situ Ti/Cr K-edge XANES) and anionic redox (in situ S K-edge XANES, Raman, XPS...) and cation migration (STEM). However, the crystallographic study needs to be revised and the presence (absence) of migrating Cr (Ti) and SS dimers formation need to be confirm by analysis of the EXAFS part of the S, Cr, Ti K-edges XAS data. Therefore, this paper should be accepted after the applying/answering the following revisions/comments:

Response:

Thank the reviewer's encouragement and supports.

Comments:

- Synthesis: The authors said that the structure of all $\text{NaCr}_{1-y}\text{Ti}_y\text{S}_2$ composition are similar to NaTiS_2 and NaCrS_2 but the diffraction pattern of NaTiS_2 looks quite different to the others.

Response:

The reviewer is right. The diffraction pattern of NaTiS_2 is different to the others. There are impurity lines of the NaTiS_2 pattern at 31° and 36° , which can be attributed to oxidized P3 structure Na_xTiS_2 . We have added these discussions to the legend of Fig. S1 as "There are impurity peaks for NaTiS_2 at 31° and 36° , which can be attributed to oxidized P3 structure Na_xTiS_2 due to the instability of Ti^{3+} and O3-structure material exposed in the air."

Comments:

- Electrochemical tests: The Columbic efficiencies at the first cycle are superior to 100% (for $y=1, 0.5$ and 0.33). Is it really possible to insert more than 1 Na^+ in $\text{Na}_x\text{Cr}_{1-y}\text{Ti}_y\text{S}_2$ materials? Na vacancies at the pristine state, often reported for the synthetize NaTMO_2 without Na precursor excess, should be the reason.

Response:

The reviewer is correct. It is impossible to insert more than 1 Na⁺ in Na_xCr_{1-y}Ti_yS₂ materials. Na₂S was not excessive because the stoichiometry mixture of Na₂S, Ti, S, and Cr powder was used for the synthesis of the NaCr_{1-y}Ti_yS₂ series materials. Na vacancies at the pristine state was formed, as often reported for NaTMO₂ synthesized, for example:

Didier C, Guignard M, Denage C, Szajwaj O, Ito S, Saadoun I, et al. Electrochemical Na-Deintercalation from NaVO₂. *Electrochemical and Solid-State Letters* 2011; **14(5):A75** According to the reviewer's suggestion, we have clarified this point in the legend of Fig. 1 as "The Columbic efficiency of NaCr_{2/3}Ti_{1/3}S₂ at the first cycle is superior to 100% for there are few Na vacancies in the pristine material without Na precursor excess."

Comments:

- in situ XAS: the author mentioned that *Cr Pre-edge intensity increases "in stage 3 that might be resulting from the local distortion of Cr-S octahedron caused by movement of S"* but an increase of Cr pre-edge is also observed during stage 1 where no distortion of S network is expected (*i.e.* no formation of SS dimers). Could you comment?

Response:

Thanks for reviewer's insightful comment.

In stage 1, there is no obvious change in the pre-edge of Cr in the potential range from 1.68 V to 1.9 V. The intensity of Cr pre-edge was increased at 2.15 V and 2.34 V and keeps the same in stage 2, because there is a little difference between the pre-edge of Cr in O3 (stage 1) and P3 (stage 2). 2.15-2.34 V is just in the process of phase transition. Thus, there is no distortion of S network in the charge potential from 1.68 V to 1.9 V in stage 1.

Comments:

o "The gradually increasing intensity of pre-edge at 4970 eV indicates the oxidation process from Ti³⁺ to Ti⁴⁺ in the first stage of charging process". The oxidation of Ti³⁺ to Ti⁴⁺ isn't supposed to lead to an increase of the pre-edge intensity as Ti³⁺ (t_{2g}¹) is Jahn Teller active whereas Ti⁴⁺ (t_{2g}⁰) isn't. Could you comment?

Response:

Thanks for the comment and we are sorry for this wrong statement. According to the reviewer's suggestion, we deleted this sentence.

Comments:

o "The pre-edge of Ti exhibited the same phenomenon as the spectra of Cr due to the comparable chemical environment of Cr-S and Ti-S". Looking the Ti K-edge data carefully, I can't see any increase of Ti K pre-edge during stage 3.

Response:

The reviewer is right. There is no increase of Ti K pre-edge during stage 3. According to the reviewer's suggestion, we have deleted this sentence.

Comments:

o "First of all, during the first stage, the S K-edge is hardly changed". The authors have assigned the first stage to Ti^{3+}/Ti^{4+} redox. If cationic redox involve in this first stage, the pre-edge intensity should increase through the generation of holes in the admixed $Ti3dS3p$. Moreover, the authors said: "In the second stage, the pre-edge located at ~ 2468 eV is gradually grown". This phenomenon is usually attributed to cationic redox via holes formation in the admixed $Ti3dS3p$. Could you comment?

Response:

Thanks for the reviewer's insightful comment. Based on the Density Functional Theory (DFT) calculation, we made the schematic energy level diagram (Figure R1). The top part of Ti^{3+} 3d orbital (i.e. energy level of Ti^{4+}/Ti^{3+} redox couple) does not overlap with S 3p orbital. This means the admixing of $Ti3dS3p$ is very weak. In the oxidation process of Ti^{3+} , the pre-edge (we have changed the description of "sulfur pre-edge" to "sulfur shoulder") of S will not rise (stage 1 in Figure 3f).

And in the second stage of charging process, because of obvious overlap of Cr 3d orbital and S 3p orbital, there is indeed admixed $Cr3dS3p$ (Figure R1). No obvious shift of K-edge can be observed in the XANES pattern of Cr and Ti in the second stage (Figure 3d-3e), indicating they hardly participate in charge compensation. Sulfur do have a rise in shoulder (stage 2 in Figure 3f), which can be derived from the formation of electron holes. So, we attribute the second stage to the formation process of electron holes on sulfur.

Figure R1. Positions of the Ti^{3+}/Ti^{2+} , Ti^{4+}/Ti^{3+} and Cr^{4+}/Cr^{3+} redox couples relative to the Fermi energy of layered $NaCr_{2/3}Ti_{1/3}S_2$.

Comments:

o "the distorting of CrS_6 octahedral structure upon the removal of Na ion should be

responsible for the formation of the typical pre-edge". The pre-edge intensity can be assigned to a distortion for TM K-edge because for those elements, the pre-edge corresponds to the $1s \rightarrow 3d$ transition which is a forbidden (due to symmetry arguments) and become allowed due to the distortion, breaking the symmetry. Whereas for $1s \rightarrow 3p$ (allowed transition) the distortion isn't supposed to have any significant effect on the pre-edge intensity.

Response:

Thanks for the reviewer's comments, we are totally in agreement with the reviewers. When talking about the "pre-edge" of Cr and Ti, it corresponds to the $1s \rightarrow 3d$ transition. When talking about the "sulfur pre-edge" of sulfur in the second stage, we actually refer to the "S shoulder" in 2468 eV, whose energy level is lower by admixing of Cr3dS3p.

In order to avoid any confusion, we have replaced "sulfur pre-edge" with "sulfur shoulder" and revised the related sentences in page 8 and 9.

Comments:

o According to your S K-edge XANES data, you conclude to the existence of S_2^{n-} but this hypothesis **should be checked with S K-edge EXAFS** part of XAS data collected. Such nice XANES (*i.e.* not noisy) data should give good enough EXAFS oscillations allowing to detect any formation of S-S dimers or S_2^{2-} species. **Moreover Cr/Ti Kedge EXAFS should give an experimental proof of their possible migration in empty Na site.**

Response:

Thanks for reviewer's valuable comments. We have supplemented the K-edge EXAFS of Cr (Figure R2), Ti(Figure R3) and S(Figure R4) in the supplementary information (Figure S10-12). We have also added related discussions as follows in the revision in page 9, 12 and 13.

In Fig. R2, there are two main peaks in the R space: the first peak at $\sim 2.40 \text{ \AA}$ corresponds to scattering of Cr-S, the second peak at $\sim 3.60 \text{ \AA}$ corresponds to scattering of Cr-Cr/Ti. It needs to be noticed that the radial distance should be added by about 0.4 \AA for phase correction to fit the XRD and DFT calculated values. The first peak representing Cr-S distance decreases in the charging process and increases in the discharging process. This is in good agreement with DFT calculation results (see Figure S26). The second peak representing nearest Cr-Cr distance, in the range $x=1.0-0.5$, decreases from 3.66 \AA to 3.45 \AA , which consists with DFT calculation. But the peak of Cr-Cr increases and broadened during the 2.9 V plateau, corresponding to x content of 0.5-0.0. Only considering the Cr-Cr distance in the same close packing layer is not enough to explain this increasement. Cr atom that migrated to Na layer can be "nearest coordination shell of Cr" as well. This kind of Cr-Cr distance is 4.394 \AA , which is meaningful in explaining scattering data. As a result of average effect, the Cr-Cr peak increases. Considering the migration ratio of 29% that drawn from STEM and DFT calculation, we can conclude that the average distance of nearest Cr-Cr is 3.72 \AA , which

consists with the EXAFS results.

Figure R2. a. *Ex situ* Cr K-edge EXAFS profile of $\text{NaCr}_{2/3}\text{Ti}_{1/3}\text{S}_2$ during the first cycle and **b.** converted to R space.

In Fig. R3b, the first peak representing Ti-S distance decreases in the charging process and increases in the discharging process, similar to Cr-S. But the intensity of Ti-(Ti/Cr) shell peak is quite low, making it difficult to be further analyzed, probably due to the low amount of Ti relative to Cr.

Figure R3. a. *Ex situ* Ti K-edge EXAFS profile of NaCr_{2/3}Ti_{1/3}S₂ during the first cycle and **b.** converted to R space.

Figure R4. a. *Ex situ* S edge EXAFS profile of NaCr_{2/3}Ti_{1/3}S₂ during the first cycle and **b.** converted to R space.

Ex situ S EXAFS during the first cycle was primitively collected, as shown in Figure R4. It can be noticed that there is an obvious peak at ~2850 eV, corresponding to the Cl in NaClO₄ electrolyte. No Cl peak can be observed in the pristine sample because it hasn't been soaked in NaClO₄ electrolyte. Other electrode samples for EXAFS measurement have been cleansed by DMC for several times, but it seems impossible to clear all the remaining ClO₄⁻, making the available range of S EXAFS shrinking to 2500-2800 eV, which is not enough to export reasonable R space profile

Then 2D X-ray fluorescence (XRF) images as well as *ex situ* S K-edge x-ray absorption spectra (XAS) were collected (see Fig. R5,6)

Figure R5. 2D scanning XRF images of S and Cl at (a-b) the pristine (immersed in the electrolyte) and (c-d) fully charged states collected at a energy of 3000 eV. The color of the each pixel in the XRF images reflects the concentration of the S and Cl elements. (dark: low concentration; yellow: high concentration)

Figure R6. Left: sulfur K-edge XANES data for pristine and full charged samples, inset is corresponding sulfur EXAFS data. Right: data converted to R space.

It can be clearly seen from the XRF images as shown in Figure R5 that the sulfur is homogeneously distributed in the pristine and charged electrodes. At the same time, the electrolyte surface was covered by Cl, which is from the NaClO_4 in the electrolyte. It should be mentioned that, the pristine electrode after immerse in the electrolyte

was used for this measurement. To avoid the interference of Cl absorption in the S- K edge XAFS part this time, a region with high sulfur concentration and low Cl was selected for the S K-edge XAS measurement (yellow and green circle in the Figure R 5 a,c). The interference of Cl was avoided which can be concluded from inset of Fig. R6 left, Cl K-edge can not be observed anymore. The change of S XANES in Fig. R6 is consistent with *in situ* results: shift of S K-edge position and rise of "S-S dimer peak" at 2470.7 eV. Furthermore, after the EXAFS data is converted to R space, an embedded peak raised at $\sim 1.6 \text{ \AA}$ can be observed. This peak can be attributed to disulfide (S-S distance 2.05 \AA) after considering phase correction, which can be a solid evidence for the formation of disulfide in full charged $\text{Na}_0\text{Cr}_{2/3}\text{Ti}_{1/3}\text{S}_2$.

A research on FeS_2 showed a similar conclusion in the position of embedded disulfide peaks (see Fig. R7, Takeuchi, T. *et al.* Preparation of $\text{Li}_2\text{S-FeS}_x$ Composite Positive Electrode Materials and Their Electrochemical Properties with Pre-Cycling Treatments. *Journal of the Electrochemical Society* **162**, A1745-A1750 (2015).)

Figure R7. The k^3 -weighted EXAFS Fourier transform magnitude for the $\text{Li}_2\text{S-FeS}$ composite ($\text{Li}_2\text{S} : \text{FeS} = 4 : 1$) positive electrodes before electrochemical tests and after the 1st and 2nd charge and discharge cycling (a) without and (b) with stepwise pre-cycling treatment. Corresponding spectra for Li_2S , S , and FeS_2 are also shown for comparison, Copyright 2015 The Electrochemical Society.

We have supplemented these conclusions in page 9 in the manuscript as well as in Fig 3 and Fig. S 6,7.

Comments:

- Rietveld refinements:

o The authors should mention that there are lines at around 11° and 14° which aren't

fitted with this model and could correspond to impurity lines or superstructure peaks associated with an eventual Cr/Ti ordering.

Response:

Thanks for reviewer's constructive comments. To elucidate where these two peaks belong, we firstly assume they were superstructure peaks. The results are shown in Fig. R8. In Fig. R8, we tried to stimulate several common supercells that lead to superstructure peaks, i.e. $1 \times 1 \times 2$, $2 \times 2 \times 2$ and $3 \times 3 \times 1$ supercells. But none of them can fit 11° and 14° peaks well.

Figure R8. XRD refinement patterns of powder $\text{NaCr}_{2/3}\text{Ti}_{1/3}\text{S}_2$ versus Bragg positions (indicated as vertical tick marks) of different supercells and Na_2SO_3 sample.

Then we turned to search for impurity substances, and found Na_2SO_3 , as a common impurity in commercial Na_2S material used in synthesis, can fit 11° and 14° peaks well. We have clarified the existence of this impurity in the legend of Fig.4a.

Comments:

o I can't imagine that the $\text{Na}(\text{Cr}_{2/3}\text{Ti}_{1/3})\text{S}_2$ and $\text{Na}_{0.67}(\text{Cr}_{2/3}\text{Ti}_{1/3})\text{S}_2$ phases crystallize in a P1 space group as mentioned in table S2 and S3. The cell parameters are very close to a hexagonal cell which is the classical space group for the O3 and P3 structures. They can sometimes be described in monoclinic symmetry when honeycomb-like superstructure peaks appear or orthorhombic cell when a Jahn Teller distortion occurs but never in triclinic cell. **The author should perform the Rietveld refinement using the appropriated space group.**

Response:

We have re-refined these two powders with R-3m and R3m space group, respectively. Although the goodness of fit for $\text{NaCr}_{2/3}\text{Ti}_{1/3}\text{S}_2$ as R-3m space groups is not enough as using P1 space group, as shown in Table R1 and Table R3, respectively, the obtained lattice parameters using R-3m and P1 space groups stay nearly the same. And there is quite slight difference for the Rietveld refinements results of $\text{Na}_{0.67}(\text{Cr}_{2/3}\text{Ti}_{1/3})\text{S}_2$ using R3m and P1 space groups (see Table R2 and Table R4).

According to the reviewer's suggestion, we have replaced the former refinements with P1 group with using typical R-3m and R3m structure in Supplementary Information.

Table R1. Structural parameters and atomic positions of as-prepared $\text{NaCr}_{2/3}\text{Ti}_{1/3}\text{S}_2$ deduced from Rietveld refinement of synchrotron data.

$\text{NaCr}_{2/3}\text{Ti}_{1/3}\text{S}_2$						
Space group		R - 3 m	Rwp=16.95%	Rp=11.41%	$\chi^2 = 1.733$	
		$a = 3.554624(20) \text{ \AA}$	$c = 19.58132(18) \text{ \AA}$	$V = 214.2690(30) \text{ \AA}^3$		
Atom	Wyckoff position	x/a	y/b	z/c	Occupancy	Uiso
Cr1	3a	0.0	0.0	0.0	0.652	0.00101(12)
Ti2	3a	0.0	0.0	0.	0.348	0.0040(14)
S3	6c	0	0	0.26519(8)	1.000	0.0032(8)
Na4	3b	0.0	0.0	0.5	0.973	0.00105(9)

Table R2. Structural parameters and atomic positions of Na_{0.66}Cr_{2/3}Ti_{1/3}S₂ deduced

from Rietveld Refinement

Na _{0.66} Cr _{2/3} Ti _{1/3} S ₂						
Space group		R 3 m	Rwp=6.45%	Rp=5.02%	x 2 = 1.208	
		a = 3.4042 (5) Å	c = 20.885 (4) Å	V = 209.60 (6) Å ³		
Atom	Wyckoff position	x/a	y/b	z/c	Occupancy	Uiso
Cr1	3a	0.0	0.0	0.0050 (22)	0.652	0.0047 (16)
Ti2	3a	0.0	0.0	0.0050 (34)	0.348	0.024 (10)
S3	3a	0	0	0.4049 (18)	1.000	0.0124 (33)
S4	9b	1/3	2/3	0.2814 (12)	1.000	0.001 (28)
Na5	3a	0.0	0.0	0.1704 (27)	0.65 (5)	0.09 (4)

Table R3. Structural parameters and atomic positions of as-preparedNaCr_{2/3}Ti_{1/3}S₂ deduced from Rietveld refinement of synchrotron data.

NaCr _{2/3} Ti _{1/3} S ₂						
Space group		P 1	Rwp=12.91%	Rp=8.24%	x 2 = 0.2047	
		a = 3.553057 (30) Å	b = 3.556203 (23) Å	c = 19.58331 (13) Å	α = 89.9878 (9) °	
		β = 89.9800 (8) °	γ = 120.07650 (30) °	V = 214.1260 (20) Å ³		
Atom	Wyckoff position	x/a	y/b	z/c	Occupancy	Uiso
Cr1	1a	0.051 (12)	0.031 (25)	-0.0025 (16)	0.624 (7)	0.0102 (25)
Cr2	1a	0.6311 (28)	0.322 (7)	0.3414 (4)	0.643 (7)	0.0209 (22)
Cr3	1a	0.2712 (31)	0.647 (8)	0.6705 (5)	0.656 (7)	0.0116 (23)
Na4	1a	0.000 (5)	0.010 (13)	0.5016 (10)	0.995 (15)	0.0107 (34)
Na5	1a	0.594 (4)	0.304 (9)	0.8390 (7)	0.988 (15)	0.013 (4)
Na6	1a	0.315 (5)	0.664 (13)	0.1714 (7)	0.923 (14)	0.0016 (35)
S7	1a	0.0027 (33)	0.004 (6)	0.2684 (4)	0.995 (18)	0.0005 (30)
S8	1a	0.6685 (35)	0.342 (7)	0.5954 (5)	0.974 (18)	0.0045 (31)
S9	1a	0.322 (4)	0.670 (7)	0.9295 (4)	0.967 (16)	0.0045 (30)
S10	1a	-0.021 (4)	-0.002 (6)	0.73690 (33)	0.992 (19)	0.0019 (34)

S11	1a	0.657(4)	0.340(6)	0.06760(34)	0.987(20)	0.0009(32)
S12	1a	0.350(4)	0.681(6)	0.3999(4)	0.953(18)	0.0053(34)
Ti13	1a	– 0.0912(33)	–0.033(9)	0.0093(5)	0.366(7)	0.049(4)
Ti14	1a	0.740(6)	0.375(13)	0.3234(8)	0.305(8)	0.047(5)
Ti15	1a	0.424(7)	0.713(13)	0.6620(8)	0.353(8)	0.031(5)

Table R4. Structural parameters and atomic positions of $\text{Na}_{0.66}\text{Cr}_{2/3}\text{Ti}_{1/3}\text{S}_2$ deduced from Rietveld Refinement

$\text{Na}_{0.66}\text{Cr}_{2/3}\text{Ti}_{1/3}\text{S}_2$						
Space group $P 1$ Rwp=10.07% Rp=8.02% $\chi^2 = 1.171$						
$a = 3.3987(7)\text{Å}$ $b = 3.3969(7)\text{Å}$ $c = 20.6597(18)\text{Å}$ $\alpha = 90.249(16)$						
$\beta = 89.693(15)$ $\gamma = 119.663(8)^\circ$ $V = 207.26(6)\text{Å}^3$						
Atom	Wyckoff position	x/a	y/b	z/c	Occupancy	Uiso
Cr1	1a	0	0	0	0.62(10)	0.09(9)
Cr2	1a	0.6666	0.3333	0.3333	0.60(5)	0.08(4)
Cr3	1a	0.3333	0.6666	0.6666	0.67(9)	0.09(7)
Na4	1a	0.0000	0.0000	0.1790	0.66(11)	0.09(7)
Na5	1a	0.6666	0.3333	0.5123	0.62(4)	0.090(22)
Na6	1a	0.3333	0.6666	0.8457	0.73(11)	0.01(4)
S7	1a	0.0000	0.0000	0.3930	0.93(12)	0.02(5)
S8	1a	0.6666	0.3333	0.7263	0.98(4)	0.09(4)
S9	1a	0.3333	0.6666	0.0597	0.91(8)	0.090(31)
S10	1a	0.0000	0.0000	0.6063	0.99(6)	0.008(28)
S11	1a	0.6666	0.3333	0.9396	0.98(6)	0.090(19)
S12	1a	0.3333	0.6666	0.2730	0.99(7)	0.058(29)
Ti13	1a	0	0	0	0.39(4)	0.09(9)
Ti14	1a	0.6666	0.3333	0.3333	0.38(6)	0.08(4)
Ti15	1a	0.3333	0.6666	0.6666	0.38(5)	0.09(7)

Comments:

o χ^2 can't be <1, it means that the model fits the noise due to the refinement of too many parameters compared to the number of experimental information available (using P1 instead of R-3m space group should be the reason of such low χ^2 value).

Response:

The reviewer is right. We have used R-3m and R3m space group instead of P1 space group for refinements with $\chi^2 > 1$.

Comments:

o The Wyckoff position are also wrong, using P1 space group, there is no symmetry elements so only 1 Wyckoff position possible: the 1a.

Response:

Thanks for your kind suggestion. We have corrected wrong Wyckoff positions (see Table R3 and Table R4) and have replaced P1 space group with R-3m and R3m space groups as shown in Table R1 and R2.

Comments:

o It's not possible to refine Ti/Cr partial occupancies based on XRD data. There are only 2 electrons difference, too low to be refined: partial occupancies need be fixed according to the composition suggested by chemical analysis.

Response:

According to the reviewer's suggestion, the composition analysis measured by ICP has been added in Table R5 and in the Supplementary Table 5, which is in agreement with the chemical stoichiometry of the mixture of Ti and Cr powder.

Table R5. Stoichiometry of as-prepared $\text{NaCr}_{2/3}\text{Ti}_{1/3}\text{S}_2$ determined from ICP analysis.

Composition from ICP			
Atom	Na	Cr	Ti
Content	0.973	0.652	0.348

Comments:

- STEM:

o The utilization of STEM to identify some Cr in the Na layer could be inconclusive. There are obvious stacking faults (occurring along the 001 direction but also along

other directions, as evidenced by the obvious anisotropic broadening of the diffraction lines seen on the XRD pattern of the full charged sample) mitigating the contrast between TM layer and “empty” Na layer. Based on STEM, the authors estimate that 25-30% of Cr migrate in empty Na site, **such huge amount of antisite defects should be seen thanks to Cr K-edge EXAFS. Moreover Ti K-edge EXAFS should also give the experimental proof the absence of Ti migration that the authors claim.**

Response:

Thanks for reviewer’s constructive suggestions.

The bright points in “empty Na layer” in Fig. 5c can be identified as migrated Cr, not stacking faults, because of their position right in the center of S octahedron.

As shown in Fig. R2 and mentioned above, the antisite of Cr in the full charged sample can be supported by Cr K-edge EXAFS.

Due to the quite low intensity of Ti-Ti/Cr shell peak, it is difficult to provide the information about Ti migration by Ti K-edge EXAFS.

Comments:

o The authors claim that the formation of the SS bond is associated to the Cr migration in its neighboring. But, according to the Fig 5, the SS distances don’t seem to depend on the amount of migrating Cr. Moreover the “short SS distance”(0.2 nm) that authors claimed to detect looks like noise: it’s probably an overinterpretation of the data.

Response:

Thanks for the reviewer’s suggestion. In order to avoid the overinterpretation of the STEM data, we have deleted the bar of (0.2 nm/0.3 nm) in Fig. 5c. The sentence “Some of S-S distance near the cation defects caused by Cr migrated to Na vacancy can reach 0.2 nm, consists with typical S-S distance in S_2^{2-} ” is also deleted.

Comments:

- Raman:

o The corresponding capacity should be mentioned on the ex-situ Raman figure (it’s difficult to compare with the electrochemistry)

Response:

Thanks for your suggestion, we have labelled corresponding charging state in the *ex situ* Raman figure. We replaced the former figure with Fig. R9.

Figure R9. *Ex situ* Raman spectrum data of $\text{NaCr}_{2/3}\text{Ti}_{1/3}\text{S}_2$ during the first cycle

Comments:

o The peak assigned to S-S dimers in $\text{Na}_x(\text{Cr,Ti})\text{S}_2$ looks higher for sample #3 than #4 and #5 (end of charge), whereas according to your theory the SS dimers are supposed to be formed only from the $x \sim 0.45$, approximately the composition of the sample #3.

Response:

Thanks for the reviewer's comment. Firstly, the intensity of Raman peak is a relative value, rougher the surface is, the higher the peaks would be, the intensity of peaks may not mean much in *ex situ* tests. Secondly the longitudinal distribution of *ex situ* sample is not uniform, the surface part might be ahead of the average charging process, which would be captured in Raman spectrum. So we described the Raman as "a new peak at about 487 cm^{-1} arose in the charging process and decreased in discharging process" to point out there are indeed S-S bonds formed in the charging process, mostly in the plateau of 2.9 V part. We have added these discussions in the Supplementary Figure 13 according to the reviewer's suggestion.

Comments:

o FeS_2 is supposed to have SS bonds but doesn't show any contribution in the "SS dimers" region. Could you comment?

Response:

Thanks for your comment. In order to avoid the confusion, we have clarified the "SS dimers" region as shown in Figure R6 according to the reviewer's suggestion.

Comments:

- General comment: Please be careful about the terms employed and the references chosen

o Line 57-58: speaking about Li-Rich NMC: "Oxygen anion redox with the reversible

formation and decomposition of O–O dimers is witnessed^{17,20,21}.” Those references refer to Li_2IrO_3 , Li_2RuO_3 and Na_3RuO_4 and not to Li-Rich NMC.

Response:

Thanks for reviewer’s comment.

Line 57-58: We have made this sentence clear with pointing out the specific materials.

The sentence “Oxygen anion redox with the reversible formation and decomposition of O–O dimers is witnessed^{17,20,21}” has been replaced with “The reversible formation and decomposition of O–O dimers can also be witnessed in Li_2IrO_3 , Li_2RuO_3 and Na_3RuO_4 ^{17,20,21}.”

Comments:

o Line 198/199: “O-O species formed in $\text{Na}_x[\text{Mg}_{0.28}\text{Mn}_{0.72}]\text{O}_2$ ²⁵”. The authors of this paper never mentioned any formation of O-O species in this compound.

Response:

We actually cited this paper as “no O-O species formed in $\text{Na}_x[\text{Mg}_{0.28}\text{Mn}_{0.72}]\text{O}_2$ ”, which we believe is consistent with this paper.

Comments:

o Lines 387/389 : “In our case, S^{2-}/S^{n-} (eq.1) and $S^{n-}/(S_2)^{n-}$ (eq.2) redox processes are similar to two types of oxygen redoxes in $\text{Li}_x\text{M}_y\text{O}_z$ reported previously, e.g. O^{n-} in $\text{Li}_{1.2}\text{Ni}_{0.13}\text{Co}_{0.13}\text{Mn}_{0.54}\text{O}_2$ and $(\text{O}_2)^{n-}$ in Li_2IrO_3 ”. That’s true but it’s in disagreement with the comment made at Line 57/58

Response:

According to the reviewer’s suggestion, we have revised the sentence in Line 57/58. “Oxygen anion redox with the reversible formation and decomposition of O–O dimers and O holes is witnessed^{17,20,21}” have been replaced with “The reversible formation of O–O dimers as well as electron holes on O can also be confirmed in Li_2IrO_3 , Li_2RuO_3 , Na_3RuO_4 ^{17,20,21} and $\text{Li}_{1.2}\text{Ni}_{0.13}\text{Co}_{0.13}\text{Mn}_{0.54}\text{O}_2$ ¹⁶, respectively.”

Comments:

In conclusion this article should be accepted after applying the revision proposed especially regarding the Rietveld refinements and treatment of the EXAFS part of the S/Cr/Ti K-edges XAS. Moreover the authors shouldn’t over-interpret the data (*i.e.* some STEM data) and need to clarify discrepancies (*i.e.* SS dimers formation seen by *ex-situ* Raman vs. state of charge).

Response:

Thanks for the reviewer’s kind comments, we have re-refined these two powders with R-3m and R3m space group instead of P1 group;

We have supplemented the K-edge EXAFS of S/Cr/Ti K-edges XAS and related

discussions;

We have deleted the related sentences which the reviewer pointed out about the overinterpretation of the STEM data.

We have revised Supplementary Figure 13 (Raman data) to figure out ex-situ Raman spectrum vs. state of charge.

Thank the reviewers for giving us the opportunity to revise this manuscript.

Reviewer #2 (Remarks to the Author):

Comments:

The authors investigated anionic redox in layered $\text{NaCr}_{2/3}\text{Ti}_{1/3}\text{S}_2$. Structural characterization was carried out to elucidate S-S formation and the mechanism of anionic redox for the active material. There was a suite of characterization techniques including XRD, XAS, HAADF-STEM, Raman, XPS, and electronic structure calculations. While this study employs many techniques, the reviewer is not convinced of dimerized S-S formation from sulfide species in the pristine material. Below are questions for clarifying some of the data supporting dimerization of S-S.

Response:

Thanks for the reviewer's comments.

We would like to clarify all the data supporting dimerization of S-S for the reviewer:

The formation of S-S dimers is directly confirmed in a model "layered $\text{Na}_x\text{Cr}_{2/3}\text{Ti}_{1/3}\text{S}_2$ " by *in situ* XAS, *ex situ* STEM, Raman, and DFT results.

In situ XAS: a new peak appears at 2470.7 eV can be clearly observed on the third stage of charging process. It corresponds to the formation of S_2^{2-} .

ex situ STEM: the shortening of the S-S distance to around 0.293 nm is witnessed by the STEM. This is a direct evidence for the formation of S-S dimers in the full charged structure.

Raman: a new peak at about 487 cm^{-1} arose in the charge process. It corresponds to the formation of S-S dimers

DFT calculation: S-S dimer can be formed between layers in the full charged structure with the lowest total energy.

It is worth mentioning that we have supplemented the sulfur K-edge EXAFS in Fig R6 as an key evidence for the existence of S-S dimers in full charged sample.

Thus, our data provide solid evidence on the formation of S-S dimers.

Comments:

Anionic dimer formation was supported by X-ray absorption spectroscopy. However, the plots are difficult to digest. While Stage 1 in Figure 3 shows no sulfide oxidation state change as evidenced by the lack of change in the S K-edge, Stage 3 depicts quite a bit of change in the post-edge region but not much in the pre-edge region. Is the sulfur dimerization occurring without changes in the oxidation state? The XAS data seems to support structural rearrangement. and not the S(-1.92 charge) to S_2 with 3.33 charge per disulfide.

Response:

Thanks for the reviewer's comments. The sulfur dimerization indeed occurs with changes in the oxidation state. A detailed explanation for the third stage in charging process of S EXAFS is shown in Fig. R10. By comparing the two spectra at the beginning and the end of the third stage separately, we can find that the "pre-edge" or the more accurate statement "shoulder" at 2468 eV disappears, indicating the decrease of sulfur electron holes because of dimerization. The main peak also moves toward high energy,

proving the oxidation process of sulfur.

Figure R10. *In situ* Cr K-edge EXAFS profile of the starting and the end in stage 3.

Comments:

The XPS data to suggest formation of oxidized forms of S (e.g. sulfide and sulfites) is depicted in supplementary Figure 9 (bottom pane). However, the fitting of pristine sulfide (S^{2-}) in $NaCr_{2/3}Ti_{1/3}S_2$ is quite poor. The $2p_{3/2}$ and $2p_{1/2}$ peaks are not well resolved for S 2p binding energies. Also, there is consistent notation that is awkward and unconventional (e.g. $2p_3$ vs. $2p_{3/2}$).

Response:

Thanks for the reviewer's correction. We have corrected the unconventional $2p_3/2p_1$ notations with $2p_{3/2}$ and $2p_{1/2}$ in Supplementary Figure 14 and 18. The fitting of pristine $NaCr_{2/3}Ti_{1/3}S_2$ has also been revised as shown in Fig. R11. $\sum \chi^2$ of the pristine sample is 2.495 now.

Figure R11. S 2p XPS data of pristine, charged and discharged sample of $\text{NaCr}_{2/3}\text{Ti}_{1/3}\text{S}_2$.

Comments:

The S-S formation was also reasoned to occur as indicated by Raman peak at 487 cm^{-1} . However, as has been demonstrated with Li-S (from S catholyte) and Na-S reports, polysulfides will form and possess normal modes over a range of $400\text{-}600\text{ cm}^{-1}$.

Response:

“According to previous reports on Li_2S_x ($x \geq 2$) sulfides, the S-S dimers show a Raman shift in the range of $400\text{-}600\text{ cm}^{-1}$ ⁴⁰. We have added this description to the Raman part according to the reviewer’s suggestion. In addition, we have clarified “S-S dimer in $\text{NaCr}_{2/3}\text{Ti}_{1/3}\text{S}_2$ region” and “possible S-S dimer range” region in different colors as shown in Figure. R6.

Reference 40 is:

Wu, H. L., Huff, L. A. & Gewirth, A. A. In situ Raman spectroscopy of sulfur speciation in lithium-sulfur batteries. *ACS Appl Mater Interfaces* **7**, 1709-1719, (2015)

Comments:

What is the mechanism of disproportionation of sulfide species near the crack in the electrode to produce both S^{2-} species (also, what are these species) and elemental sulfur when desodiating from $Na=0.5$ to $Na=0$ in $Na(x)Cr(2/3)Ti(1/3)S_2$?

Response:

The mechanism of disproportionation of sulfide species near the full charged $Na_0Cr_{2/3}Ti_{1/3}S_2$ crack is proposed to be

This chemical equilibrium can move to the right with the precipitation of elemental sulfur near the crack, on the surface. This mechanism is raised to provide one possible origin of elemental sulfur. The main course of sulfur precipitation should come from the further oxidation of $(S_2)^{2-}$ in the charging process. These discussions have been added in the legend of Supplementary Figure S17.

Reviewer #3 (Remarks to the Author):

Comments:

This paper reports anionic redox processes in sulfide electrode $\text{NaCr}_2/3\text{Ti}_1/3\text{S}_2$ for Na ion batteries and their extensive characterization with most modern and sophisticated physical-chemical methods, with emphasis on structural characterization. This is a hot topic, with many papers being published on sodium ion batteries every year. The authors have done a substantial amount of work on synthesis and characterization of their model electrode material, accumulated extensive data, analyzed them carefully and observed some interesting trends revealing the importance of anionic processes. I think that the paper is suitable for publication after minor revision. My comments are listed below:

Response:

Thanks for the reviewer's encouragement and comments for this paper.

Comments:

1) Do you present electrochemical measurements for only one coin cell? Have you tried to assemble several ones, and if so, how reproducible were the results?

Response:

We present electrochemical measurements for not only one coin cell. More than tens of coin cells were assembled to test the electrochemical measurements, and more than 50 test cells were assembled to perform all kinds of characterizations with at least one charging process. After the experimental conditions are settled, the repeatability of electrochemical performance becomes quite steady. Their capacities can reach at least 180 ± 10 mAh/g in the first cycle, which we used in the manuscript. We have added this information into the legend of Supplementary Figure S4 as the reviewer's suggestion.

Comments:

2) In the abstract, you mention that the capacity is 190 mAh/g, but this number is only achieved in the first cycle. For Li-ion batteries, the 1st discharge-charge cycle is often referred to as a "formation cycle", shouldn't you put in the Abstract the capacity value, which is more accurately reflects the real performance? Or, the data in Figure 1 represents actually the data for capacity retention after 50 cycles based on the second cycle discharge capacity, that is, after the formation cycle?

Response:

Thanks for reviewer's constructive suggestions. We have corrected the expression in the abstract and the electrochemical performance part. We replaced the first cycle capacity with the reversible capacity of the second cycle (186 mAh/g, 0.95 Na per unit). The data in Figure 1 represents actually the data for capacity retention after 50 cycles based on the first cycle. In order to avoid the confusion, we have added this

information into the legend of Supplementary Figure S5 as the reviewer's suggestion.

Comments:

3) In the Abstract, the expression "anion redox reaction between S^{2-} and $(S_2)^{2-}$ " is confusing, as it sounds like these two groups interact with each other, while in fact they transform to each other.

Response:

Thanks for meticulous checking. According to the reviewer's suggestion, "Although the anion redox reaction between S^{2-} and $(S_2)^{2-}$ through dimerization of S-S in sulfides" has been replaced with "Although the anion redox couple of S^{2-} and $(S_2)^{2-}$ through dimerization of S-S in sulfides".

Comments:

The sentence "In this study, various charge compensation mechanisms of sulfur anionic redox process in layered $NaCr_{2/3}Ti_{1/3}S_2$ through the formation of electron holes, disulfide-like species, the precipitation of elemental sulfur, as well as S-S dimerization, especially through the formation of electron holes are investigated" seems to be somewhat redundant.

"formation of ... disulfide-like species" and "S-S dimerization" are the same?

"formation of electron holes"... "especially through the formation of electron holes"?.. mentioned twice

Response:

Thanks for pointing out. We have revised this sentence and deleted the first "electron holes".

The sentence "through the formation of electron holes, disulfide-like species, the precipitation of elemental sulfur, as well as S-S dimerization, especially through the formation of electron holes are investigated" has been replaced with "through the formation of disulfide-like species, the precipitation of elemental sulfur, as well as S-S dimerization, especially through the formation of electron holes are investigated".

Based on previous literatures, the definition of different chalcogens are listed here:

Based on Luo, K. *et al.* Charge-compensation in 3d-transition-metal-oxide intercalation cathodes through the generation of localized electron holes on oxygen. *Nat Chem* **8**, 684-691, doi:10.1038/nchem.2471 (2016).

$(O_2)^{n-}$ (O–O distance 2.42 Å) is called **$(O_2)^{n-}$ species**

O_2^{2-} , (O–O distance ~1.45 Å) **peroxide or peroxide species**

Based on Mccalla, E. *et al.* Visualization of O-O peroxo-like dimers in high-capacity layered oxides for Li-ion batteries. *Science* **350**, 1516 (2015).

$(O_2)^{n-}$ (O–O distance 2.45 Å) **peroxo-like species or peroxo species or O-O dimers.**

O_2^{2-} , (O–O distance~1.5 Å) **peroxide species**

Based on Rong, X. *et al.* Structure-Induced Reversible Anionic Redox Activity in Na Layered Oxide Cathode. *Joule* **2**, 125-140, doi:10.1016/j.joule.2017.10.008 (2018). $(O_2)^{n-}$ (O–O distance 2.5 Å) is called **peroxo-like species or peroxo-like O-O dimer** O_2^{2-} , (O–O distance ~1.5 Å) peroxide or **peroxide pair**

Many other similar references are omitted here. It can be concluded that the name of O_2^{2-} (O–O ~1.5 Å), the real peroxide, is directly called peroxide species. So in sulfur systems, S_2^{2-} (S–S ~2.1 Å) can be called disulfides or disulfide-like species. The word “dimer” is only used as the name of $(O_2)^{n-}$, so we use dimer to describe $(S_2)^{n-}$ (S–S ~2.93 Å). Our purpose is to unify the name of sulfides and oxides, make them easier to understand, easier to be associated with other chalcogenides, which may help to link the nature of anion redox processes. The definition of all these species are listed in equations of the Discussion part, and we have made a detailed definition after the four equations to make them clear.

It is worth mentioning that we have supplemented the sulfur K-edge EXAFS in Fig R6 as an key evidence for the existence of S-S dimers in full charged sample.

Comments:

4) As you discuss the advantages of the anionic redox processes, it would be helpful to comment, how your mixed-metal material compares to other polysulfide electrodes operating due to redox in disulfide group $(S-S)_2^- + 2e^- = 2S_2^-$ (for example, TiS_3). As Nature communications is a multidisciplinary journal, for a wider audience, it would be helpful to put your work in a more general context.

Response:

Thanks for the reviewer’s suggestion. We have summarized the performance of various polysulfide electrodes materials for lithium or sodium ion batteries in Table R6 (Table S2 in the manuscript). After comparison, $NaCr_{2/3}Ti_{1/3}S_2$ may potentially provide a higher reversibility to other transition-metal polysulfides such as TiS_3 , FeS_2 , VS_4 and TiS_4 [23-28 in the manuscript]. The higher reversibility of $NaCr_{2/3}Ti_{1/3}S_2$ may thanks to its stable layered metal–sulfur framework and highly reversible nature of localized electron holes. To balance the capacity and capacity retention in metal sulfides or metal polysulfides, more works should be done in the future to fully explore the potential for secondary batteries, and we wish to provide a new approach and a deep insight of sulfur redox to extend the development prospects for metal sulfide electrodes.

According to the reviewer’ s suggestion, we have added these discussions in the revision:

“Comparing to other metal polysulfide electrodes²³⁻²⁸ for lithium or sodium batteries (Table S2), it is clearly shown that poor cycling stability is a common problem to be solved in metal sulfide electrodes driven by anion redox reactions, to which the highly reversibility of electron holes might be a potential solution as we discuss later.”

Table R6. Summary of various polysulfide electrodes materials for lithium or sodium ion batteries.

Materials	Electrochemical window (V)	Capacity (mAh g ⁻¹)	Cycling stability	References
Bulk NaCr _{2/3} Ti _{1/3} S ₂	1.4-3.3	186	52.2 % (360 cycles)	This work
Na ₁₅ Sn ₄ /Na ₃ PS ₄ glass-ceramic/a-TiS ₃ .	1.2-2.6	300	33 % (10 cycles)	Journal of Power Sources 275 , 284-287,2015
Na ₁₅ Sn ₄ /Na ₃ PS ₄ glass-ceramic/a-TiS ₃ with acetylene black	1.2-2.6	350	85.7 % (5 cycles)	Journal of Power Sources 275 , 284-287,2015
FeS ₂ +Li ₂ S blended powder	1.0-3.0	650	26 % (15 cycles)	J Electrochem Soc 159 , A75-A84 (2012).
FeS ₂ -Li ₂ S composite	1.0-3.0	790	64 % (15 cycles)	J Electrochem Soc 159 , A75-A84 (2012).
VS ₄ /rGO composite versus Na ⁺ /Na	0.01–2.2	450.4	53.5 % (50 cycles)	Acs Appl Mater Inter 7 , 20902-20908 (2015).
VS ₄ /rGO composite versus Li ⁺ /Li	0.01–3.0	1669	57.2% (100 cycles)	J Am Chem Soc 135 , 8720-8725, (2013).
a-TiS ₄ versus Li ⁺ /Li	1.6-3.0	609	67.1% (20 cycles)	J Am Chem Soc 139 , 8796-8799 (2017).

Comments:

5) Figure 5 c for the fully charged sample implies that the Cr migration phenomenon is rather pronounced (29 % (25 % by DFT) as stated in the paper), so, do you have a correspondingly significant number of S atoms paired in dimers in the fully charged sample? So, not only S atoms at the edges of the layers form dimers? Fig. S18 is not completely clear.

Response:

We do have a correspondingly significant number of S atoms paired in dimers in the fully charged sample. It depends on the ratio of migrated Cr. And we drew this

conclusion from DFT calculation results. When the ratio of Cr migration is 0, there are no S-S bond formed in the relaxed structure. When the ratio of Cr migration is 25%, 1/3 of S formed S-S dimers (Fig. S23 left). When the ratio of Cr migration is 50%, 1/6 of S formed S-S dimers (Fig. S23 right). Not only S atoms at the edges of the layers form dimers, the dimers are formed between layers in the crystal bulk. If analyzed by average oxidation state, the oxidation of sulfur of fully charged $\text{Na}_0\text{Cr}_{2/3}\text{Ti}_{1/3}\text{S}_2$ is -1.66 (as we further discussed in comment 7#), which can be considered as 1/3 of sulfur in -1 valence to form S-S dimers and 2/3 of sulfur in -2 valence. This ratio is consistent with DFT calculation results with 25% Cr migration. We have added the number of S atoms paired in dimers in the legend of Fig. S23.

Comments:

The terms “dimers” and “disulfides” seem to be used in different meanings, which is a bit confusing and should be defined.

Response:

Thanks for your kind comment, as we discussed above the definitions have been made clear. The sentence “We define S-S dimer as $(\text{S}_2)^{n-}$ (broadly speaking, $2 \leq n < 4$) here, the disulfides are defined as exactly $(\text{S}_2)^{2-}$, which can be considered as a special case of S-S dimer.” was added to the discuss part in the manuscript.

Comments:

6) Please specify more clearly, whether the 4 equations in the Discussion represent 4 consecutive steps? If so, than in the Abstract the phrases like “other anion redox mechanism” are inaccurate, as this is not another mechanism, but just a step in a more complex process.

Response:

Thanks for your suggestion, the 4 equations in the discussion are indeed 4 consecutive steps from S^{2-} to element S. So we have revised this sentence in the Abstract to avoid misunderstanding and to clarify the formation of electron hole is one of the intermediate processes.

“however, other anion redox mechanism such as through electron hole formation has not been investigated.” is replaced by “however, anion redox process such as through electron hole formation has not been investigated.”

Comments:

7) How did you obtain the numbers 3.33 and 3.84 for charge in the Eq. (2)?

Response:

Thanks for your comment. These numbers are obtained from the Na content in the charging process. Firstly in the stage 2 of charging process, the electron hole on $\text{S}^{1.92-}$ is formed. The number 1.92 was obtained from sodium removal amount in stage 2, in

which the $\text{Na}_{0.66}\text{Cr}_{2/3}\text{Ti}_{1/3}\text{S}_2$ was oxidized to $\text{Na}_{0.5}\text{Cr}_{2/3}\text{Ti}_{1/3}\text{S}_2$ under a desodiation process. We consider S^{2-} as the major electron donor in this stage as we analyzed in XAS part, S give 0.16 electron per $\text{Na}_x\text{Cr}_{2/3}\text{Ti}_{1/3}\text{S}_2$ unit. So the average valence state of S rises from S^{2-} to $\text{S}^{1.92-}$. Then $\text{S}_2^{3.33-}$ will eventually formed at the end of charging process as we described in Eq. (2). We assume in this process sulfur is the major electron donor as well, sulfur give 0.66 electron per $\text{Na}_x\text{Cr}_{2/3}\text{Ti}_{1/3}\text{S}_2$ unit in total. Then number 3.33 can be obtained via calculating $(-2 \times 2 + 0.66)$

Comments:

8) Recent reviews on anionic redox in electrode materials should be cited: Grayfer, E. D. et al. (2017). Anionic redox chemistry in polysulfide electrode materials for rechargeable batteries. *ChemSusChem*, 10(24), 4805-4811.
Zhao, C. et al. (2017). Review on anionic redox for high-capacity lithium-and sodium-ion batteries. *Journal of Physics D: Applied Physics*, 50(18), 183001.
Yang, W., & Devereaux, T. P. (2018). Anionic and cationic redox and interfaces in batteries: Advances from soft X-ray absorption spectroscopy to resonant inelastic scattering. *Journal of Power Sources*, 389, 188-197.

Response:

Thanks for your kind suggestion. These references (References 8, 23 and 12, respectively) have been cited in the revision.

Comments:

9) At some (not numerous) points, grammar/typos should be revised, like “possible routes of anionic redox processes from the electron holes to element is” – routes are “The localized electron holes (S^n) is formed” are formed
“the capacity from ~100 mAh/g in NaCrS_2 to ~190 mAh/g.” sounds like incomplete sentence
“Sample preparation. To synthesize the $\text{NaCr}_{1-y}\text{Ti}_y\text{S}_2$ serious materials” – series?
“the mixture of Na_2S , Ti, S, and Cr powder in stoichiometry was compressed” - which stoichiometry?

Response:

Thanks for reviewer’s advice. We sincerely apologize for our mistakes and we have corrected them as follows in the manuscript.

Page 5, line 74: routes of anionic redox processes from the electron holes to element is → are

Page 18, line 371: localized electron holes (S^n) is formed → are formed

Page 19, line 394: “from ~100 mAh/g in NaCrS_2 to ~190 mAh/g” →
“from ~100 mAh/g in NaCrS_2 to ~186 mAh/g in this system”

Page 21, line 414: serious → series

Page 21, line 415: in stoichiometry → in a stoichiometry of 3:2:9:4

In brief, all concerns from the referees have been clarified and all suggestions have been taken. We believe that the revised manuscript can satisfy the requirements from the publication in *Nature Communications*. Your positive responses are appreciated very much. If you have further questions, please contact us.

Best regards,
Sincerely yours
Zhengwen Fu

The changes and responses to the reviewers are listed below point by point:

Reviewer #1 (Remarks to the Author):

Review of « Anionic redox reaction in layered $\text{NaCr}_{2/3}\text{Ti}_{1/3}\text{S}_2$ through electron holes formation in addition to the dimerization of S-S»

Comments:

The authors studied an O3- $\text{NaCr}_{2/3}\text{Ti}_{1/3}\text{S}_2$ cathode, delivering a capacity beyond the $\text{Ti}^{3+}/\text{Ti}^{4+}$ redox limit based on combined cationic and anionic redox processes. Various charge compensation mechanisms involving sulfur have been studied/discussed such as formation of localized holes ($\text{S}^{\cdot-}$), S_2^{2-} species, S-S dimers ($\text{S}_2^{\cdot-}$), and dissolution of elemental Sulfur in the electrolyte. The authors used a wide range of relevant characterization techniques to characterize the cationic (in situ Ti/Cr K-edge XANES) and anionic redox (in situ S K-edge XANES, Raman, XPS...) and cation migration (STEM). However, the crystallographic study needs to be revised and the presence (absence) of migrating Cr (Ti) and SS dimers formation need to be confirm by analysis of the EXAFS part of the S, Cr, Ti K-edges XAS data. Therefore, this paper should be accepted after the applying/answering the following revisions/comments:

Response:

Thank the reviewer's encouragement and supports.

Comments:

- Synthesis: The authors said that the structure of all $\text{NaCr}_{1-y}\text{Ti}_y\text{S}_2$ composition are similar to NaTiS_2 and NaCrS_2 but the diffraction pattern of NaTiS_2 looks quite different to the others.

Response:

The reviewer is right. The diffraction pattern of NaTiS_2 is different to the others. There are impurity lines of the NaTiS_2 pattern at 31° and 36° , which can be attributed to oxidized P3 structure Na_xTiS_2 . We have added these discussions to the legend of Fig. S1 as "There are impurity peaks for NaTiS_2 at 31° and 36° , which can be attributed to oxidized P3 structure Na_xTiS_2 due to the instability of Ti^{3+} and O3-structure material exposed in the air."

Reviewer response:

OK

Comments:

- Electrochemical tests: The Columbic efficiencies at the first cycle are superior to 100% (for $y=1, 0.5$ and 0.33). Is it really possible to insert more than 1 Na^+ in $\text{Na}_x\text{Cr}_{1-y}\text{Ti}_y\text{S}_2$ materials? Na vacancies at the pristine state, often reported for the synthetize

NaTMO₂ without Na precursor excess, should be the reason.

Response:

The reviewer is correct. It is impossible to insert more than 1 Na⁺ in Na_xCr_{1-y}Ti_yS₂ materials. Na₂S was not excessive because the stoichiometry mixture of Na₂S, Ti, S, and Cr powder was used for the synthesis of the NaCr_{1-y}Ti_yS₂ series materials. Na vacancies at the pristine state was formed, as often reported for NaTMO₂ synthesized, for example:

Didier C, Guignard M, Denage C, Szajwaj O, Ito S, Saadouni I, et al. Electrochemical Na-Deintercalation from NaVO₂. *Electrochemical and Solid-State Letters* 2011; **14(5):A75**

According to the reviewer's suggestion, we have clarified this point in the legend of Fig. 1 as "The Columbic efficiency of NaCr_{2/3}Ti_{1/3}S₂ at the first cycle is superior to 100% for there are few Na vacancies in the pristine material without Na precursor excess."

Reviewer response:

So could you provide the ICP results in order to discriminate between Na vacancies at the pristine state which are filled during the discharge or the formation of any SEI consuming some electrons during discharge?

Comments:

- in situ XAS: the author mentioned that *Cr Pre-edge intensity increases "in stage 3 that might be resulting from the local distortion of Cr-S octahedron caused by movement of S"* but an increase of Cr pre-edge is also observed during stage 1 where no distortion of S network is expected (*i.e.* no formation of SS dimers). Could you comment?

Response:

Thanks for reviewer's insightful comment.

In stage 1, there is no obvious change in the pre-edge of Cr in the potential range from 1.68 V to 1.9 V. The intensity of Cr pre-edge was increased at 2.15 V and 2.34 V and keeps the same in stage 2, because there is a little difference between the pre-edge of Cr in O3 (stage 1) and P3 (stage 2). 2.15-2.34 V is just in the process of phase transition. Thus, there is no distortion of S network in the charge potential from 1.68 V to 1.9 V in stage 1.

Reviewer response:

OK

Comments:

o "The gradually increasing intensity of pre-edge at 4970 eV indicates the oxidation process from Ti³⁺ to Ti⁴⁺ in the first stage of charging process". The oxidation of Ti³⁺ to Ti⁴⁺ isn't supposed to lead to an increase of the pre-edge intensity as Ti³⁺ (t_{2g}¹) is Jahn Teller active whereas Ti⁴⁺ (t_{2g}⁰) isn't. Could you comment?

Response:

Thanks for the comment and we are sorry for this wrong statement. According to the reviewer's suggestion, we deleted this sentence.

Reviewer response:

OK

Comments:

o *"The pre-edge of Ti exhibited the same phenomenon as the spectra of Cr due to the comparable chemical environment of Cr-S and Ti-S"*. Looking the Ti K-edge data carefully, I can't see any increase of Ti K pre-edge during stage 3.

Response:

The reviewer is right. There is no increasement of Ti K pre-edge during stage 3. According to the reviewer's suggestion, we have deleted this sentence.

Reviewer response:

OK

Comments:

o *"First of all, during the first stage, the S K-edge is hardly changed"*. The authors have assigned the first stage to Ti^{3+}/Ti^{4+} redox. If cationic redox involve in this first stage, the pre-edge intensity should increase through the generation of holes in the admixed $Ti3dS3p$. Moreover, the authors said: *"In the second stage, the pre-edge located at ~2468 eV is gradually grown"*. This phenomenon is usually attributed to cationic redox via holes formation in the admixed $Ti3dS3p$. Could you comment?

Response:

Thanks for the reviewer's insightful comment. Based on the Density Functional Theory (DFT) calculation, we made the schematic energy level diagram (Figure R1). The top part of Ti^{3+} 3d orbital (i.e. energy level of Ti^{4+}/Ti^{3+} redox couple) does not overlap with S 3p orbital. This means the admixing of $Ti3dS3p$ is very weak. In the oxidation process of Ti^{3+} , the pre-edge (we have changed the description of "sulfur pre-edge" to "sulfur shoulder") of S will not rise (stage 1 in Figure 3f).

And in the second stage of charging process, because of obvious overlap of Cr 3d orbital and S 3p orbital, there is indeed admixed $Cr3dS3p$ (Figure R1). No obvious shift of K-edge can be observed in the XANES pattern of Cr and Ti in the second stage (Figure 3d-3e), indicating they hardly participate in charge compensation. Sulfur do have a rise in shoulder (stage 2 in Figure 3f), which can be derived from the formation of electron holes. So, we attribute the second stage to the formation process of electron holes on sulfur.

Figure R1. Positions of the $\text{Ti}^{3+}/\text{Ti}^{2+}$, $\text{Ti}^{4+}/\text{Ti}^{3+}$ and $\text{Cr}^{4+}/\text{Cr}^{3+}$ redox couples relative to the Fermi energy of layered $\text{NaCr}_{2/3}\text{Ti}_{1/3}\text{S}_2$.

Reviewer response:

I can't believe in a such ionic Ti-S bond. Looking at the Ti K-edge data, the pre-edge increases while Ti is oxidized. This indicates that the Ti-S bond isn't purely ionic and therefore, the oxidation of Ti should generate holes in the $\text{Ti}3\text{d}3\text{p}$ orbitals and should lead to an increasing in the S pre-edge.

Comments:

o "the distorting of CrS_6 octahedral structure upon the removal of Na ion should be responsible for the formation of the typical pre-edge". The pre-edge intensity can be assigned to a distortion for TM K-edge because for those elements, the pre-edge corresponds to the $1s \rightarrow 3d$ transition which is a forbidden (due to symmetry arguments) and become allowed due to the distortion, breaking the symmetry. Whereas for $1s \rightarrow 3p$ (allowed transition) the distortion isn't supposed to have any significant effect on the pre-edge intensity.

Response:

Thanks for the reviewer's comments, we are totally in agreement with the reviewers. When talking about the "pre-edge" of Cr and Ti, it corresponds to the $1s \rightarrow 3d$ transition. When talking about the "sulfur pre-edge" of sulfur in the second stage, we actually refer to the "S shoulder" in 2468 eV, whose energy level is lower by admixing of $\text{Cr}3\text{d}3\text{p}$.

In order to avoid any confusion, we have replaced "sulfur pre-edge" with "sulfur shoulder" and revised the related sentences in page 8 and 9.

Reviewer response:

OK

Comments:

o According to your S K-edge XANES data, you conclude to the existence of S_2^{n-} but this hypothesis **should be checked with S K-edge EXAFS** part of XAS data collected. Such nice XANES (*i.e.* not noisy) data should give good enough EXAFS oscillations allowing to detect any formation of S-S dimers or S_2^{2-} species. **Moreover Cr/Ti K-edge EXAFS should give an experimental proof of their possible migration in empty Na site.**

Response:

Thanks for reviewer's valuable comments. We have supplemented the K-edge EXAFS of Cr (Figure R2), Ti(Figure R3) and S(Figure R4) in the supplementary information (Figure S10-12). We have also added related discussions as follows in the revision in page 9, 12 and 13.

In Fig. R2, there are two main peaks in the R space: the first peak at $\sim 2.40 \text{ \AA}$ corresponds to scattering of Cr-S, the second peak at $\sim 3.60 \text{ \AA}$ corresponds to scattering of Cr-Cr/Ti. It needs to be noticed that the radial distance should be added by about 0.4 \AA for phase correction to fit the XRD and DFT calculated values. The first peak representing Cr-S distance decreases in the charging process and increases in the discharging process. This is in good agreement with DFT calculation results (see Figure S26). The second peak representing nearest Cr-Cr distance, in the range $x=1.0-0.5$, decreases from 3.66 \AA to 3.45 \AA , which consists with DFT calculation. But the peak of Cr-Cr increases and broadened during the 2.9 V plateau, corresponding to x content of $0.5-0.0$. Only considering the Cr-Cr distance in the same close packing layer is not enough to explain this increasement. Cr atom that migrated to Na layer can be "nearest coordination shell of Cr" as well. This kind of Cr-Cr distance is 4.394 \AA , which is meaningful in explaining scattering data. As a result of average effect, the Cr-Cr peak increases. Considering the migration ratio of 29% that drawn from STEM and DFT calculation, we can conclude that the average distance of nearest Cr-Cr is 3.72 \AA , which consists with the EXAFS results.

Figure R2. a. *Ex situ* Cr K-edge EXAFS profile of NaCr_{2/3}Ti_{1/3}S₂ during the first cycle and **b.** converted to R space.

In Fig. R3b, the first peak representing Ti-S distance decreases in the charging process and increases in the discharging process, similar to Cr-S. But the intensity of Ti-(Ti/Cr) shell peak is quite low, making it difficult to be further analyzed, probably due to the low amount of Ti relative to Cr.

Figure R3. a. *Ex situ* Ti K-edge EXAFS profile of NaCr_{2/3}Ti_{1/3}S₂ during the first cycle and **b.** converted to R space.

Figure R4. a. *Ex situ* S edge EXAFS profile of $\text{NaCr}_{2/3}\text{Ti}_{1/3}\text{S}_2$ during the first cycle and **b.** converted to R space.

Ex situ S EXAFS during the first cycle was primitively collected, as shown in Figure R4. It can be noticed that there is an obvious peak at ~ 2850 eV, corresponding to the Cl in NaClO_4 electrolyte. No Cl peak can be observed in the pristine sample because it hasn't been soaked in NaClO_4 electrolyte. Other electrode samples for EXAFS measurement have been cleansed by DMC for several times, but it seems impossible to clear all the remaining ClO_4^- , making the available range of S EXAFS shrinking to 2500-2800 eV, which is not enough to export reasonable R space profile

Then 2D X-ray fluorescence (XRF) images as well as *ex situ* S K-edge x-ray absorption spectra (XAS) were collected (see Fig. R5,6)

Figure R5. 2D scanning XRF images of S and Cl at (a-b) the pristine (immersed in the electrolyte) and (c-d) fully charged states collected at a energy of 3000 eV. The color of the each pixel in the XRF images reflects the concentration of the S and Cl elements. (dark: low concentration; yellow: high concentration)

Figure R6. Left: sulfur K-edge XANES data for pristine and full charged samples, inset is corresponding sulfur EXAFS data. Right: data converted to R space.

It can be clearly seen from the XRF images as shown in Figure R5 that the sulfur is homogeneously distributed in the pristine and charged electrodes. At the same time, the electrolyte surface was covered by Cl, which is from the NaClO₄ in the electrolyte. It should be mentioned that, the pristine electrode after immerse in the electrolyte

was used for this measurement. To avoid the interference of Cl absorption in the S- K edge XAFS part this time, a region with high sulfur concentration and low Cl was selected for the S K-edge XAS measurement (yellow and green circle in the Figure R 5 a,c). The interference of Cl was avoided which can be concluded from inset of Fig. R6 left, Cl K-edge can not be observed anymore. The change of S XANES in Fig. R6 is consistent with *in situ* results: shift of S K-edge position and rise of "S-S dimer peak" at 2470.7 eV. Furthermore, after the EXAFS data is converted to R space, an embedded peak raised at $\sim 1.6 \text{ \AA}$ can be observed. This peak can be attributed to disulfide (S-S distance 2.05 \AA) after considering phase correction, which can be a solid evidence for the formation of disulfide in full charged $\text{Na}_0\text{Cr}_{2/3}\text{Ti}_{1/3}\text{S}_2$.

A research on FeS_2 showed a similar conclusion in the position of embedded disulfide peaks (see Fig. R7, Takeuchi, T. *et al.* Preparation of Li_2S - FeS_x Composite Positive Electrode Materials and Their Electrochemical Properties with Pre-Cycling Treatments. *Journal of the Electrochemical Society* **162**, A1745-A1750 (2015).)

Figure R7. The k^3 -weighted EXAFS Fourier transform magnitude for the Li_2S - FeS composite ($\text{Li}_2\text{S} : \text{FeS} = 4 : 1$) positive electrodes before electrochemical tests and after the 1st and 2nd charge and discharge cycling (a) without and (b) with stepwise pre-cycling treatment. Corresponding spectra for Li_2S , S, and FeS_2 are also shown for comparison, Copyright 2015 The Electrochemical Society.

We have supplemented these conclusions in page 9 in the manuscript as well as in Fig 3 and Fig. S 6,7.

Reviewer response:

Cr K-edge: The Cr-M distance is probably not the best feature to look at, the Cr-M peak shape/intensity should be more meaningful. Indeed, if Cr migrates from the TM layer

to the AM layer, vacancies should be generated in the TM layer therefore you should see a decrease in intensity in the main peak of the 2nd shell (here at *ca.* 3.2Å) as well as the appearance of a shoulder at higher distance (or a new peak, depending the resolution) associated to the presence Cr in the AM layer. This combination of features isn't observed in the data provided.

Ti K-edge: Something must be wrong the data extraction. In EXAFS every signal below 1Å aren't real features but everything above should be (if the data extraction is done properly). Therefore, the peak at 1.5Å is supposed to be real. For the pristine sample, it could correspond to JT activity of Ti³⁺ but in that case this peak should disappear at the charged states, where Ti is its 4+ state and not JT active anymore. Moreover, the Cr migration could also be evidenced at Ti K-edge though similar features as those described above for Cr K-edge EXAFS.

S K-edge: Really nice work. I really appreciate the efforts made by the authors to suppress the contribution of Cl in the S K-edge and provide nice data. However, the full story (*i.e.* S K-edge EXAFS data at the end of stage 1 and 2 in addition to pristine and fully charged) would be even more convincing and would allow to associate electronic states (XANES) to structural features (EXAFS).

Comments:

- Rietveld refinements:

o The authors should mention that there are lines at around 11° and 14° which aren't fitted with this model and could corresponds to impurity lines or superstructure peaks associated with an eventual Cr/Ti ordering.

Response:

Thanks for reviewer's constructive comments. To elucidate where these two peaks belong, we firstly assume they were superstructure peaks. The results are shown in Fig. R8. In Fig. R8, we tried to stimulate several common supercells that lead to superstructure peaks, *i.e.* 1×1×2, 2×2×2 and 3×3×1 supercells. But none of them can fit 11° and 14° peaks well.

Figure R8. XRD refinement patterns of powder $\text{NaCr}_{2/3}\text{Ti}_{1/3}\text{S}_2$ versus Bragg positions (indicated as vertical tick marks) of different supercells and Na_2SO_3 sample.

Then we turned to search for impurity substances, and found Na_2SO_3 , as a common impurity in commercial Na_2S material used in synthesis, can fit 11° and 14° peaks well. We have clarified the existence of this impurity in the legend of Fig.4a.

Reviewer response:

OK

Comments:

o I can't imagine that the $\text{Na}(\text{Cr}_{2/3}\text{Ti}_{1/3})\text{S}_2$ and $\text{Na}_{0.67}(\text{Cr}_{2/3}\text{Ti}_{1/3})\text{S}_2$ phases crystallize in a P1 space group as mentioned in table S2 and S3. The cell parameters are very close a hexagonal cell which is the classical space group for the O3 and P3 structures. They can sometime be described in monoclinic symmetry when honeycomb-like superstructure peaks appear or orthorhombic cell when a Jahn Teller distortion occurs but never in triclinic cell. **The author should perform the Rietveld refinement using the appropriated space group.**

Response:

We have re-refined these two powders with R-3m and R3m space group, respectively. Although the goodness of fit for $\text{NaCr}_{2/3}\text{Ti}_{1/3}\text{S}_2$ as R-3m space groups is not enough as using P1 space group, as shown in Table R1 and Table R3, respectively, the obtained lattice parameters using R-3m and P1 space groups stay nearly the same. And there is quite slight difference for the Rietveld refinements results of $\text{Na}_{0.67}(\text{Cr}_{2/3}\text{Ti}_{1/3})\text{S}_2$ using R3m and P1 space groups (see Table R2 and Table R4).

According to the reviewer's suggestion, we have replaced the former refinements with P1 group with using typical R-3m and R3m structure in Supplementary Information.

Table R1. Structural parameters and atomic positions of as-prepared $\text{NaCr}_{2/3}\text{Ti}_{1/3}\text{S}_2$ deduced from Rietveld refinement of synchrotron data.

$\text{NaCr}_{2/3}\text{Ti}_{1/3}\text{S}_2$						
Space group		$R - 3m$	$R_{wp}=16.95\%$		$R_p=11.41\%$	$\chi^2 = 1.733$
$a = 3.554624(20) \text{ \AA}$		$c = 19.58132(18) \text{ \AA}$		$V = 214.2690(30) \text{ \AA}^3$		
Atom	Wyckoff position	x/a	y/b	z/c	Occupancy	Uiso
Cr1	3a	0.0	0.0	0.0	0.652	0.00101(12)
Ti2	3a	0.0	0.0	0.	0.348	0.0040(14)
S3	6c	0	0	0.26519(8)	1.000	0.0032(8)
Na4	3b	0.0	0.0	0.5	0.973	0.00105(9)

Table R2. Structural parameters and atomic positions of $\text{Na}_{0.66}\text{Cr}_{2/3}\text{Ti}_{1/3}\text{S}_2$ deduced from Rietveld Refinement

$\text{Na}_{0.66}\text{Cr}_{2/3}\text{Ti}_{1/3}\text{S}_2$						
Space group		R 3 m	Rwp=6.45%	Rp=5.02%	$\chi^2 = 1.208$	
		$a = 3.4042(5) \text{ \AA}$	$c = 20.885(4) \text{ \AA}$	$V = 209.60(6) \text{ \AA}^3$		
Atom	Wyckoff position	x/a	y/b	z/c	Occupancy	Uiso
Cr1	3a	0.0	0.0	0.0050(22)	0.652	0.0047(16)
Ti2	3a	0.0	0.0	0.0050(34)	0.348	0.024(10)
S3	3a	0	0	0.4049(18)	1.000	0.0124(33)
S4	9b	1/3	2/3	0.2814(12)	1.000	0.001(28)
Na5	3a	0.0	0.0	0.1704(27)	0.65(5)	0.09(4)

Table R3. Structural parameters and atomic positions of as-prepared $\text{NaCr}_{2/3}\text{Ti}_{1/3}\text{S}_2$ deduced from Rietveld refinement of synchrotron data.

$\text{NaCr}_{2/3}\text{Ti}_{1/3}\text{S}_2$						
Space group		P 1	Rwp=12.91%	Rp=8.24%	$\chi^2 = 0.2047$	
		$a = 3.553057(30) \text{ \AA}$	$b = 3.556203(23) \text{ \AA}$	$c = 19.58331(13) \text{ \AA}$	$\alpha = 89.9878(9)^\circ$	
		$\beta = 89.9800(8)^\circ$		$\gamma = 120.07650(30)^\circ$	$V = 214.1260(20) \text{ \AA}^3$	
Atom	Wyckoff position	x/a	y/b	z/c	Occupancy	Uiso
Cr1	1a	0.051(12)	0.031(25)	-0.0025(16)	0.624(7)	0.0102(25)
Cr2	1a	0.6311(28)	0.322(7)	0.3414(4)	0.643(7)	0.0209(22)
Cr3	1a	0.2712(31)	0.647(8)	0.6705(5)	0.656(7)	0.0116(23)
Na4	1a	0.000(5)	0.010(13)	0.5016(10)	0.995(15)	0.0107(34)
Na5	1a	0.594(4)	0.304(9)	0.8390(7)	0.988(15)	0.013(4)
Na6	1a	0.315(5)	0.664(13)	0.1714(7)	0.923(14)	0.0016(35)
S7	1a	0.0027(33)	0.004(6)	0.2684(4)	0.995(18)	0.0005(30)
S8	1a	0.6685(35)	0.342(7)	0.5954(5)	0.974(18)	0.0045(31)
S9	1a	0.322(4)	0.670(7)	0.9295(4)	0.967(16)	0.0045(30)
S10	1a	-0.021(4)	-0.002(6)	0.73690(33)	0.992(19)	0.0019(34)
S11	1a	0.657(4)	0.340(6)	0.06760(34)	0.987(20)	0.0009(32)
S12	1a	0.350(4)	0.681(6)	0.3999(4)	0.953(18)	0.0053(34)
Ti13	1a	-	-0.033(9)	0.0093(5)	0.366(7)	0.049(4)
		0.0912(33)				

Ti14	1a	0.740(6)	0.375(13)	0.3234(8)	0.305(8)	0.047(5)
Ti15	1a	0.424(7)	0.713(13)	0.6620(8)	0.353(8)	0.031(5)

Table R4. Structural parameters and atomic positions of Na_{0.66}Cr_{2/3}Ti_{1/3}S₂ deduced from Rietveld Refinement

Na _{0.66} Cr _{2/3} Ti _{1/3} S ₂						
Space group P 1 Rwp=10.07% Rp=8.02% $\chi^2 = 1.171$						
a = 3.3987(7)Å b = 3.3969(7)Å c = 20.6597(18) Å $\alpha = 90.249(16)$						
$\beta = 89.693(15)$ $\gamma = 119.663(8)^\circ$ V = 207.26(6) Å ³						
Atom	Wyckoff position	x/a	y/b	z/c	Occupancy	Uiso
Cr1	1a	0	0	0	0.62(10)	0.09(9)
Cr2	1a	0.6666	0.3333	0.3333	0.60(5)	0.08(4)
Cr3	1a	0.3333	0.6666	0.6666	0.67(9)	0.09(7)
Na4	1a	0.0000	0.0000	0.1790	0.66(11)	0.09(7)
Na5	1a	0.6666	0.3333	0.5123	0.62(4)	0.090(22)
Na6	1a	0.3333	0.6666	0.8457	0.73(11)	0.01(4)
S7	1a	0.0000	0.0000	0.3930	0.93(12)	0.02(5)
S8	1a	0.6666	0.3333	0.7263	0.98(4)	0.09(4)
S9	1a	0.3333	0.6666	0.0597	0.91(8)	0.090(31)
S10	1a	0.0000	0.0000	0.6063	0.99(6)	0.008(28)
S11	1a	0.6666	0.3333	0.9396	0.98(6)	0.090(19)
S12	1a	0.3333	0.6666	0.2730	0.99(7)	0.058(29)
Ti13	1a	0	0	0	0.39(4)	0.09(9)
Ti14	1a	0.6666	0.3333	0.3333	0.38(6)	0.08(4)
Ti15	1a	0.3333	0.6666	0.6666	0.38(5)	0.09(7)

Reviewer response:

The GoF will always be better using a P1 space group instead of another, even the good one. P1 S.G doesn't evolve any symmetry elements therefore the absence of systematic extension and the large degree of freedom for atomic position allow the perfect fit of the data with the wrong model...

Comments:

χ^2 can't be <1, it means that the model fits the noise due to the refinement of too many parameters compared to the number of experimental information available (using P1 instead of R-3m space group should be the reason of such low χ^2 value).

Response:

The reviewer is right. We have used R-3m and R3m space group instead of P1 space group for refinements with $\chi^2 > 1$.

Reviewer response:

OK

Comments:

The Wyckoff position are also wrong, using P1 space group, there is no symmetry elements so only 1 Wyckoff position possible: the 1a.

Response:

Thanks for your kind suggestion. We have corrected wrong Wyckoff positions (see Table R3 and Table R4) and have replaced P1 space group with R-3m and R3m space groups as shown in Table R1 and R2.

Reviewer response:

OK

Comments:

It's not possible to refine Ti/Cr partial occupancies based on XRD data. There are only 2 electrons difference, too low to be refined: partial occupancies need be fixed according to the composition suggested by chemical analysis.

Response:

According to the reviewer's suggestion, the composition analysis measured by ICP has been added in Table R5 and in the Supplementary Table 5, which is in agreement with the chemical stoichiometry of the mixture of Ti and Cr powder.

Table R5. Stoichiometry of as-prepared $\text{NaCr}_{2/3}\text{Ti}_{1/3}\text{S}_2$ determined from ICP analysis.

Composition from ICP			
Atom	Na	Cr	Ti
Content	0.973	0.652	0.348

Reviewer response:

Na and S occupancies have be refined

Comments:

- STEM:

o The utilization of STEM to identify some Cr in the Na layer could be inconclusive. There are obvious stacking faults (occurring along the 001 direction but also along other directions, as evidenced by the obvious anisotropic broadening of the diffraction lines seen on the XRD pattern of the full charged sample) mitigating the contrast between TM layer and “empty” Na layer. Based on STEM, the authors estimate that 25-30% of Cr migrate in empty Na site, **such huge amount of antisite defects should be seen thanks to Cr K-edge EXAFS. Moreover Ti K-edge EXAFS should also give the experimental proof the absence of Ti migration that the authors claim.**

Response:

Thanks for reviewer’s constructive suggestions.

The bright points in “empty Na layer” in Fig. 5c can be identified as migrated Cr, not stacking faults, because of their position right in the center of S octahedron.

As shown in Fig. R2 and mentioned above, the antisite of Cr in the full charged sample can be supported by Cr K-edge EXAFS.

Due to the quite low intensity of Ti-Ti/Cr shell peak, it is difficult to provide the information about Ti migration by Ti K-edge EXAFS.

Reviewer response:

Stacking faults should also generate bright spots right in the center of S octahedron. Moreover, based on the EXFAS data provided, I’m not really convince by the TM migration that the authors claim (see comments above)

Comments:

o The authors claim that the formation of the SS bond is associated to the Cr migration in its neighboring. But, according to the Fig 5, the SS distances don’t seem to depend on the amount of migrating Cr. Moreover the “short SS distance”(0.2 nm) that authors claimed to detect looks like noise: it’s probably an overinterpretation of the data.

Response:

Thanks for the reviewer’s suggestion. In order to avoid the overinterpretation of the STEM data, we have deleted the bar of (0.2 nm/0.3 nm) in Fig. 5c. The sentence “Some of S-S distance near the cation defects caused by Cr migrated to Na vacancy can reach 0.2 nm, consists with typical S-S distance in S_2^{2-} ” is also deleted.

Reviewer response:

OK

Comments:

- Raman:

o The corresponding capacity should be mentioned on the ex-situ Raman figure (it's difficult to compare with the electrochemistry)

Response:

Thanks for your suggestion, we have labelled corresponding charging state in the *ex situ* Raman figure. We replaced the former figure with Fig. R9.

Figure R9. *Ex situ* Raman spectrum data of NaCr_{2/3}Ti_{1/3}S₂ during the first cycle

Reviewer response:

OK

Comments:

o The peak assigned to S-S dimers in Na_x(Cr,Ti)S₂ looks higher for sample #3 than #4 and #5 (end of charge), whereas according to your theory the SS dimers are supposed to be formed only from the x~0.45, approximately the composition of the sample#3.

Response:

Thanks for the reviewer's comment. Firstly, the intensity of Raman peak is a relative value, rougher the surface is, the higher the peaks would be, the intensity of peaks may not mean much in *ex situ* tests. Secondly the longitudinal distribution of *ex situ* sample is not uniform, the surface part might be ahead of the average charging process, which would be captured in Raman spectrum. So we described the Raman as "a new peak at about 487 cm⁻¹ arose in the charging process and decreased in discharging process" to point out there are indeed S-S bonds formed in the charging process, mostly in the plateau of 2.9 V part. We have added these discussions in the Supplementary Figure 13 according to the reviewer's suggestion.

Reviewer response:

OK

Comments:

o FeS₂ is supposed to have SS bonds but doesn't show any contribution in the "SS dimers" region. Could you comment?

Response:

Thanks for your comment. In order to avoid the confusion, we have clarified the "SS dimers" region as shown in Figure R6 according to the reviewer's suggestion.

Reviewer response:

OK

Comments:

- General comment: Please be careful about the terms employed and the references chosen

o Line 57-58: speaking about Li-Rich NMC: "*Oxygen anion redox with the reversible formation and decomposition of O–O dimers is witnessed*^{17,20,21}." Those references refer to Li₂IrO₃, Li₂RuO₃ and Na₃RuO₄ and not to Li-Rich NMC.

Response:

Thanks for reviewer's comment.

Line 57-58: We have made this sentence clear with pointing out the specific materials.

The sentence "Oxygen anion redox with the reversible formation and decomposition of O–O dimers is witnessed^{17,20,21}" has been replaced with "The reversible formation and decomposition of O–O dimers can also be witnessed in Li₂IrO₃, Li₂RuO₃ and Na₃RuO₄^{17,20,21},"

Reviewer response:

OK

Comments:

o Line 198/199: "*O-O species formed in Na_x[Mg_{0.28}Mn_{0.72}]O₂²⁵*". The authors of this paper never mentioned any formation of O-O species in this compound.

Response:

We actually cited this paper as "**no** O-O species formed in Na_x[Mg_{0.28}Mn_{0.72}]O₂", which we believe is consistent with this paper.

Reviewer response:

OK

Comments:

o Lines 387/389 :“In our case, S^{2-}/S^{n-} (eq.1) and $S^{n-}/(S_2)^{n-}$ (eq.2) redox processes are similar to two types of oxygen redoxes in $Li_xM_yO_z$ reported previously, e.g. O^{n-} in $Li_{1.2}Ni_{0.13}Co_{0.13}Mn_{0.54}O_2$ and $(O_2)^{n-}$ in Li_2IrO_3 ”. That’s true but it’s in disagreement with the comment made at Line 57/58

Response:

According to the reviewer’s suggestion, we have revised the sentence in Line 57/58. “Oxygen anion redox with the reversible formation and decomposition of O–O dimers and O holes is witnessed^{17,20,21}” have been replaced with “The reversible formation of O–O dimers as well as electron holes on O can also be confirmed in Li_2IrO_3 , Li_2RuO_3 , Na_3RuO_4 ^{17,20,21} and $Li_{1.2}Ni_{0.13}Co_{0.13}Mn_{0.54}O_2$ ¹⁶, respectively.”

Reviewer response:

OK

Comments:

In conclusion this article should be accepted after applying the revision proposed especially regarding the Rietveld refinements and treatment of the EXAFS part of the S/Cr/Ti K-edges XAS. Moreover the authors shouldn’t over-interpret the data (*i.e.* some STEM data) and need to clarify discrepancies (*i.e.* SS dimers formation seen by *ex-situ* Raman vs. state of charge).

Response:

Thanks for the reviewer’s kind comments, we have re-refined these two powders with R-3m and R3m space group instead of P1 group;

We have supplemented the K-edge EXAFS of S/Cr/Ti K-edges XAS and related discussions;

We have deleted the related sentences which the reviewer pointed out about the overinterpretation of the STEM data.

We have revised Supplementary Figure 13 (Raman data) to figure out *ex-situ* Raman spectrum vs. state of charge.

Thank the reviewers for giving us the opportunity to revise this manuscript.

Reviewer response:

This new version is much more convincing regarding the S redox mechanism with the nice S K-edge EXAFS data provided. However Cr/Ti K-edge EXAFS doesn’t evidence any migration of TM in the alkali layer. This discrepancy must be clarified. This paper should be accepted after the applying the suggested revisions.

Reviewer #2 (Remarks to the Author):

Given the revisions, I recommend publication of this manuscript.

Reviewer #1 (Remarks to the Author):

Reviewer response:

So could you provide the ICP results in order to discriminate between Na vacancies at the pristine state which are filled during the discharge or the formation of any SEI consuming some electrons during discharge?

Response:

Thanks for the reviewer's advice. We have measured the ICP results of the full charged sample and full discharged sample. The results are listed in Table R1, 2. From Table R1 it can be seen that there is 0.119 mol Na⁺ per Na_xCr_{2/3}Ti_{1/3}S₂ in full charged sample, indicating the removal of sodium is not totally thorough. According to the electrochemical profile in Figure.1a, up to 0.95 mol Na⁺ per unit can be removed and the extra Na⁺ detected by ICP measurement could be resulted from the formation of Na-containing SEI or remaining NaClO₄ electrolyte.

From Table R2 it can be seen that Na vacancies are filled during the discharge process. The extra Na⁺ could be Na-containing SEI or remaining NaClO₄ electrolyte as well. We have supplemented these two tables in Supplementary Information as Table S5 and S6.

Table R1. Stoichiometry of full charged Na₀Cr_{2/3}Ti_{1/3}S₂ determined from ICP analysis.

Composition from ICP			
Atom	Na	Cr	Ti
Content	0.119	0.659	0.341

Table R2. Stoichiometry of full discharged $\text{NaCr}_{2/3}\text{Ti}_{1/3}\text{S}_2$ determined from ICP analysis.

Composition from ICP			
Atom	Na	Cr	Ti
Content	1.041	0.662	0.338

Reviewer response:

I can't believe in a such ionic Ti-S bond. Looking at the Ti K-edge data, the pre-edge increases while Ti is oxidized. This indicates that the Ti-S bond isn't purely ionic and therefore, the oxidation of Ti should generate holes in the $\text{Ti}3\text{dS}3\text{p}$ orbitals and should lead to an increasing in the S pre-edge.

Response:

The reviewer is right. Ti-S bond is mostly ionic, but there is actually a little covalency between Ti and S according to our DOS results. There is small change in S K-edge in stage 1, but not much, which may be attributed to $\text{Ti}3\text{dS}3\text{p}$ admixed orbitals. For Ti K-edge data in stage 1, the pre-edge increases while Ti is oxidized. This kind of change may be attributed to either $\text{Ti}3\text{dS}3\text{p}$ admixed orbitals or change in Ti chemical environment between O3 and P3 phase, like the rise of Cr pre-edge at the end of stage 1. We have supplemented these discussions in page 8, XAS part of the manuscript.

Reviewer response:

Cr K-edge: The Cr-M distance is probably not the best feature to look at, the Cr-M peak shape/intensity should be more meaningful. Indeed, if Cr migrates from the TM layer to the AM layer, vacancies should be generated in the TM layer therefore you should see a decrease in intensity in the main peak of the 2nd shell (here at ca. 3.2Å) as well as the appearance of a shoulder at higher distance (or a new peak, depending the resolution) associated to the presence Cr in the AM layer. This combination of features isn't observed in the data provided.

Response:

Thanks for the reviewer's suggestion. We have reorganized the charging plots of the figure without plot stacking. The trend of Cr-M peaks intensity at $\sim 3.2 \text{ \AA}$ is exactly like what the reviewer said. We have supplemented this figure and corresponding description to Supplementary Figure 10.

Figure R1. *Ex situ* Cr K-edge EXAFS profile of $\text{NaCr}_{2/3}\text{Ti}_{1/3}\text{S}_2$ during the first cycle converted to R space.

Ti K-edge:

Something must be wrong the data extraction. In EXAFS every signal below 1Å aren't real features but everything above should be (if the data extraction is done properly). Therefore, the peak at 1.5Å is supposed to be real. For the pristine sample, it could correspond to JT activity of Ti^{3+} but in that case this peak should disappear at the charged states, where Ti is its 4+ state and not JT active anymore. Moreover, the Cr migration could also be evidenced at Ti K-edge though similar features as those described above for Cr K-edge EXAFS.

Response:

The reviewer is right. Something was wrong with the previous data extraction. We have reprocessed and drawn the Ti EXAFS figure according to the reviewer's suggestion. With the new data extraction, peaks below 1.8 Å disappear, indicating they were generated from wrong data extraction. The new figure shows the Ti-S distance decreases in the charging process and increases in the discharging process. We have replaced Supplementary Figure 11 with this figure. But the intensity of Ti-(Ti/Cr) shell peak is low as the previous version, which can not show enough evidence for regular changes in peak shape or peak intensity.

Figure R2. *Ex situ* Ti K-edge EXAFS profile of $\text{NaCr}_{2/3}\text{Ti}_{1/3}\text{S}_2$ during the first cycle converted to R space.

S K-edge: Really nice work. I really appreciate the efforts made by the authors to suppress the contribution of Cl in the S K-edge and provide nice data. However, the full story (*i.e.* S K-edge EXAFS data at the end of stage 1 and 2 in addition to pristine and fully charged) would be even more convincing and would allow to associate electronic states (XANES) to structural features (EXAFS).

Response:

Thanks for the reviewer's recognition. Due to lack of synchrotron beamline operation time, we can not perform further XAS measurements. In fact, the existing S EXAFS data of the pristine and full charged sample are strong enough to prove the formation of S-S dimer in the charging process. We are glad to dig deeper in corresponding S EXAFS researches in the future.

Reviewer response:

The GoF will always be better using a P1 space group instead of another, even the good one. P1 S.G doesn't evolve any symmetry elements therefore the absence of systematic extension and the large degree of freedom for atomic position allow the perfect fit of the data with the wrong model...

Response: The reviewer is right. We have applied the correct space group model and will choose the space group carefully in future works.

Reviewer response:

Na and S occupancies have be refined

Response: Based on the reviewer's suggestion, we have fixed the contents of Na, Cr, Ti and S according to ICP results in Table. S5. Their occupancies have not be refined.

Reviewer response:

Stacking faults should also generate bright spots right in the center of S octahedron. Moreover, based on the EXFAS data provided, I'm not really convince by the TM migration that the authors claim (see comments above)

Response: Thanks for the reviewer's viewpoints. We believe the bright points are exactly migrated Cr ions. Stacking faults mostly occur within a/b plane in layered cathode materials, in most of the literatures as well as this manuscript, a/b plane instead of c plane stacking faults are observed and reported. This means if the bright points are in S layers, they could be S atom columns, but the bright spots are in the center of S octahedron, which can not be directly observed S atom columns. We have assumed another possibility, that is the bright points are virtual image generated from electron beam scattering, but based on previous reports, such as X. Rong. *et al.* Anionic Redox Reaction-Induced High Capacity and Low-Strain Cathode with Suppressed Phase Transition. *Joule* 3, 503-517 (2019). and Figure S7 in our Supplementary Information, the stacking faults will not generate virtual image in the center of S octahedron. So, we believe the bright points are exactly Cr ions migrated.

Figure R3. Local Structural Changes on the Particle Surface in the charging process of $\text{Na}_{0.72}[\text{Li}_{0.24}\text{Mn}_{0.76}]\text{O}_2$, copyright CellPress, 2019.

To prove the bright spots are globally migrated Cr in bulk, we have supplemented more STEM images of full charged sample in large-scale, as shown in Fig. R4. From Fig. R4 it can be seen that the migrated Cr ions are not localized or in small amount, but global and wide-ranging. So, we believe the bright spots are indeed migrated Cr, not stacking faults. We have supplemented this figure as Supplementary Figure 27.

Figure R4. HAADF-STEM images of full charged $\text{Na}_0\text{Cr}_{2/3}\text{Ti}_{1/3}\text{S}_2$. scale bar 5 nm.

Reviewer response:

This new version is much more convincing regarding the S redox mechanism with the nice S K-edge EXAFS data provided. However Cr/Ti K-edge EXAFS doesn't evidence any migration of TM in the alkali layer. This discrepancy must be clarified. This paper should be accepted after the applying the suggested revisions.

Response

Thanks for the reviewer's kind comments. We have supplemented more STEM images of full charged sample in large-scale, which stood as a strong evidence of TM migration instead of local stacking faults.

We have reprocessed the Ti EXAFS data and rearranged EXAFS data of Cr and Ti in R space as Fig. S10, S11, in which the former showed the evidence of TM migration. The discrepancy has been clarified.

Thank the reviewers for giving us the opportunity to revise this manuscript

Reviewer #2 (Remarks to the Author):

Given the revisions, I recommend publication of this manuscript.

Response: Thanks for the reviewer's affirmation, we appreciate the valuable comments that the reviewer gave.

Reviewer #1 (Remarks to the Author):

ICP:

Would be perfect with the standard deviations

Cr/Ti K-edge EXAFS:

Ti K-edge EXAFS seems to reveal the features associated to TM migration but Cr K-edge EXAFS doesn't, did you apply the same data extraction parameters for both edges? If not you should apply the parameters used at Ti K-edge for the Cr K-edge (similar k-range for FT, same window shape, dk and k weight...). If Cr migrates you should observe, at the Cr K-edge, a similar evolution of the 2nd shell peak shape/intensity to that observed for Ti with even more obvious changes of the Ti-M(in TM layer, 2.7Å) and Ti-M(in Alkali layer, 3Å) intensity ratio and if not Ti might migrate instead. Here, it's clearly different, the Cr K-edge 2nd shell peak completely disappears at charged state, it shows the typical evolution for an amorphisation which clearly doesn't occur in this material according to the other data that you've provided: the Cr K-edge EXAFS extraction parameters must be wrong

EXAFS being an element sensitive probe of the changes in the local environment is the most relevant technique to look at TM migration, therefore if the features mentioned above are seen in the Cr K-edge EXAFS, these nice TEM data could be explained based on the EXAFS but TEM isn't sufficient to differentiate between Ti or Cr migration, if any.

This paper should be accepted after applying the revisions proposed

:

Reviewers' comments:

Reviewer #1 (Remarks to the Author):

ICP:

Would be perfect with the standard deviations

Response:

Thanks for the reviewer's advice. We have supplemented the standard deviation (SD) and relative standard deviation (RSD) of the ICP result, as listed in Table R1, 2, 3. These SD and RSD are obtained and converted through three times of consecutive measurements on every single sample. The accuracy meets the requirement of measurement. We have replaced the former ICP Table S5, S6 and S7 with the new ones.

Table R1. Stoichiometry of pristine $\text{NaCr}_{2/3}\text{Ti}_{1/3}\text{S}_2$ determined from ICP analysis.

Composition from ICP			
Atom	Na	Cr	Ti
Content	0.973	0.652	0.348
SD	0.011	0.007	0.005
RSD (%)	1.1	1.1	1.4

Table R2. Stoichiometry of full charged $\text{Na}_0\text{Cr}_{2/3}\text{Ti}_{1/3}\text{S}_2$ determined from ICP analysis.

Composition from ICP			
Atom	Na	Cr	Ti
Content	0.119	0.659	0.341
SD	0.004	0.009	0.005
RSD (%)	3.3	1.4	1.5

Table R3. Stoichiometry of full discharged $\text{NaCr}_{2/3}\text{Ti}_{1/3}\text{S}_2$ determined from ICP analysis.

Composition from ICP			
Atom	Na	Cr	Ti
Content	1.041	0.662	0.338
SD	0.009	0.011	0.006
RSD (%)	0.9	1.7	1.8

Reviewer response:

Cr/Ti K-edge EXAFS:

Ti K-edge EXAFS seems to reveal the features associated to TM migration but Cr K-edge EXAFS doesn't, did you apply the same data extraction parameters for both edges? If not you should apply the parameters used at Ti K-edge for the Cr K-edge (similar k-range for FT, same window shape, dk and k weight...). If Cr migrates you should observe, at the Cr K-edge, a similar evolution of the 2nd shell peak shape/intensity to that observed for Ti with even more obvious changes of the Ti-M (in TM layer, 2.7Å) and Ti-M (in Alkali layer, 3Å) intensity ratio and if not Ti might migrate instead. Here, it's clearly different, the Cr K-edge 2nd shell peak completely disappears at charged state, it shows the typical evolution for an amorphisation which clearly doesn't occur in this material according to the other data that you've provided: the Cr K-edge EXAFS extraction parameters must be wrong.

EXAFS being an element sensitive probe of the changes in the local environment is the most relevant technique to look at TM migration, therefore if the features mentioned above are seen in the Cr K-edge EXAFS, these nice TEM data could be explained based on the EXAFS but TEM isn't sufficient to differentiate between Ti or Cr migration, if any.

Response:

The reviewer is right. We totally agree with that EXAFS is the most relevant technique to figure out TM migration, but several factors might influence the accuracy in practice. In our case, the noisy data in high range of k space is the prior factor that disturbed further analysis of radial distribution function. In the old versions the data extraction parameters for Ti and Cr were not always the same, as we discuss later. Here we have supplemented the *in situ* EXAFS data of Cr in the charging process, which has better data quality.

As shown in Figure R1, the *ex situ* Ti and Cr k space EXAFS profile is noisy in high range. In the first revised version of our article, the parameters for Ti and Cr K-edge EXAFS were the same. we chose the range of 3.2-13.5 \AA^{-1} in k space to perform FT, used Hanning type window and the plotting k weighs were 3, as a typically value for $N < 36$ elements. The wide k range for FT led to an obvious problem that the reviewer pointed out in the second round of review, that is "there are several non-real peaks below 1.5 \AA in Ti EXAFS R space profile, resulted from wrong data extraction (shown in Figure R2a)."

To solve this problem, we changed the range to 2.5-9.1 \AA^{-1} for Ti to perform FT in the second revised version to avoid using the noisy data. But shrinking the k range will not lead in more information, in consideration of the intensity of the second shell in Figure R2a is not high enough to obtain valuable information, the "second shell" in Figure R2b seems to be mainly from the sidelobe of first shell Ti-S peak generated in FT process. The position of the second peak is below 3 \AA , which should be 3.25 to 3 \AA considering ~ 0.4 \AA of phase correction, indicating the "second peak" is sidelobe as well. The low intensity, quality of data and relatively low content of Ti are the reasons that further analyzed of these peaks can not be performed in Ti EXAFS data.

To prove the migration of Cr through EXAFS data, we have supplemented *in situ* EXAFS data of Cr during the first charging process. These data were not shown in previous versions because they were only collected in the charging process instead of the whole charging-discharging process. As shown in Figure R1c, the data quality is high enough to be further analyzed. We chose the range of 3.3-11.4 \AA^{-1} in k space to perform FT, used Hanning type window and the plotting k weighs were 3 as well. The results were shown in Figure R3a, b. The position of the first peak decreases in the charging process and the intensity of the second peak decreased, indicating the generation of Cr vacancy and migration of Cr atoms. The second shell peaks do not completely disappear at charged state in *in situ* data, excluding the possibility of complete amorphization (but the intragranular crack may indeed generate some local amorphization region as well, as STEM images showed in Fig. 5f, g in the manuscript). This is a strong evidence for Cr migration through Cr EXAFS data, and the trend of peak evolution in *in situ* data is in consistent with the *ex situ* data shown in Figure R3c. We have supplemented related discussion in page 13 of the manuscript and supplemented the *in situ* Cr EXAFS data in Figure S10.

Figure R1. a. *ex situ* Ti and b. Cr compared to c. *in situ* Cr k space EXAFS profile of $\text{NaCr}_{2/3}\text{Ti}_{1/3}\text{S}_2$ during the first cycle

Figure R2. *ex situ* Ti EXAFS profile of $\text{NaCr}_{2/3}\text{Ti}_{1/3}\text{S}_2$ during the first cycle in R space. a. k-range $3.2\text{-}13.5 \text{ \AA}^{-1}$, b. k-range $2.5\text{-}9.1 \text{ \AA}^{-1}$

Figure R3. a,b. *in situ* Cr EXAFS profile of $\text{NaCr}_{2/3}\text{Ti}_{1/3}\text{S}_2$ during the first cycle in R space and c. corresponding *ex situ* profile.

Reviewer response:

This paper should be accepted after applying the revisions proposed

Response

Thanks for the reviewer's valuable comments. We have supplemented SD and RSD values for all ICP tables. We have supplemented data extraction details in EXAFS data and supplemented *in situ* Cr EXAFS data, which stood as a strong evidence of Cr migration.

Thank the reviewers for giving us the opportunity to revise this manuscript again. We believe that the revised manuscript can satisfy the requirements from the publication in *Nature Communications*.

In brief, all concerns from the referees have been clarified and all suggestions have been taken. We believe that the revised manuscript can satisfy the requirements from the publication in *Nature Communications*. Your positive responses are appreciated very much. If you have further questions, please contact us.

Best regards,
Sincerely yours
Zhengwen Fu

REVIEWERS' COMMENTS:

Reviewer #1 (Remarks to the Author):

Given the revisions, I recommend publication of this manuscript.